# Citizen science for predicting spatio-temporal patterns in seabird abundance during migration

**Beatriz Martín**[1]*, **Alejandro Onrubia**[1], **Julio González-Arias**[2‡], **Juan A. Vicente-Vírseda**[2‡]

**1** Fundación Migres, CIMA, Tarifa, Spain, **2** Business and Finance Department, Faculty of Economics and Business, UNED, Madrid, Spain

☯ These authors contributed equally to this work.
‡ These authors also contributed equally to this work.
* bmartin@fundacionmigres.org

**Data Availability Statement:** Regarding the data used in this study, Trektellen is a public database of migration / seawatch counts and ringing results

## Abstract

Pelagic seabirds are elusive species which are difficult to observe, thus determining their spatial distribution during the migration period is a difficult task. Here we undertook the first long-term study on the distribution of migrating shearwaters from data gathered within the framework of citizen science projects. Specifically, we collected daily abundance (only abundance given presence) of Balearic shearwaters from 2005 to 2017 from the online databases Trektellen and eBird. We applied machine-learning techniques, specifically Random Forest regression models, to predict shearwater abundance during migration using 15 environmental predictors. We built separated models for pre-breeding and post-breeding migration. When evaluated for the total data sample, the models explained more than 52% of the variation in shearwater abundance. The models also showed good ability to predict shearwater distributions for both migration periods (correlation between observed and predicted abundance was about 70%). However, relative variable importance and variation among the models built with different training data subsamples differed between migration periods. Our results showed that data gathered in citizen science initiatives together with recently available high-resolution satellite imagery, can be successfully applied to describe the migratory spatio-temporal patterns of seabird species accurately. We show that a predictive modelling approach may offer a powerful and cost-effective tool for the long-term monitoring of the migratory patterns in sensitive marine species, as well as to identify at sea areas relevant for their protection. Modelling approaches can also be essential tools to detect the impacts of climate and other global changes in this and other species within the range of the training data.

## Introduction

Many seabirds are upper trophic level consumers that can be used as indicators of the status and change of pelagic ecosystems [1]. Seabirds are also relevant species from a conservation

(https://www.trektellen.nl/). Trektellen database can be accessed searching by location, date and particular species using the website tools. All the environmental predictors used in the present study are freely available from the websites of the different data sources indicated in Table 1, as well as from Trektellen website. Data sourced by eBird can be obtained after registration and request at https://ebird.org/. For further information, please see https://ebird.org/science/download-ebird-data-products.

**Funding:** This study is part of a research project ("Environmental factors determining the interannual variation in the migration of Balearic and Scopoli's shearwaters in the Mediterranean", 2018-2019) which has been partly financed by the Annual Programme of Grants of the Instituto de Estudios Ceutíes (IEC, Autonomous City of Ceuta, Spain), years 2018-2019. All the funding and sources of support (whether external or internal to our organization) received during this study have been reported and there was no additional external funding received for this study.

**Competing interests:** The authors have declared that no competing interests exist.

perspective. They are one of the most threatened groups of marine vertebrates, partly because they are highly mobile. Migratory seabirds are particularly susceptible to a large number of stressors, given the variety of habitats they use throughout their year-cycle [2]. Studies on migration and at sea distribution of pelagic seabirds have received significant attention in recent years, but there is still a gap in the knowledge on migration and at sea distribution in many of these species [3]. As other migratory species, seabirds may stopover during the journey, although the exact location of these intermediate steps is frequently unknown [4]. Stopovers are key sites and conditions experienced by seabirds in this areas can affect individual survival undermining the population size [3]. Therefore, conservation efforts should address not only wintering and breeding grounds, but the en-route locations during migration [5].

However, pelagic seabirds are elusive species which are difficult to observe, thus determining their spatial distribution during the migration period is a difficult task [4, 6]. Ringing recoveries and ocean sightings are useful for identifying very general movement patterns, but they are not enough to determine stopover areas, which are of main importance for understanding the migratory patterns as well as to inform conservation planning at sea [7, 8]. The continuous technical development of electronic devices for tracking animal movements (such as geolocators, GPSs, and PTT devices) have contributed to the knowledge on the at-sea distribution of many seabird species [9, 10]. However, even with good sample sizes, tracking data may not represent the full species migratory range due to the variety of migratory strategies at colony- and individual levels [7, 8, 11]. Therefore, important stopover areas and foraging grounds for a given species may not be identified. This is particularly true for long-lived species, such as many seabirds, which have a great capacity to alter migratory behavior in response to environmental variability [7] and in relation to specific individual traits such as age, sex or breeding colony [12, 13]. Consequently, electronic devices, even providing very detailed information on individual seabird's movements, have a limited ability to improve our understanding of the adaptation of migration strategies to deal with a changing environment at both population and species levels [14, 15]. In contrast, census methods do not allow to detect birds when they use areas out of human sight, but they can provide a valuable overall picture despite the missing information on the age and the colony where the bird breeds.

In this sense, the active public involvement in scientific research (i.e., citizen science) has become a key source of high-quality data for scientists and policymakers [16]. Among others, citizen science projects have enabled researchers to obtain a comprehensive picture of habitat use in many different species [17, 18]. These citizen science projects provide millions of species observations each year [19, 20]. But there are concerns regarding the scientific use of citizen science data [21]. For instance, data collected by volunteers is assumed to be less accurate than data collected professionally, although the few studies that compare the precision of volunteer and professional data did not conclusively show this fact [22]. However, it seems clear that, without proper standardized survey protocols and volunteer training, volunteer data may be highly variable in terms of precision (e.g., errors in species identification or biases in count estimates, uneven sampling effort, both in terms of temporal and spatial coverage, among others). In contrast to these caveats, as compared with the detailed data gathered from electronic devices, the massive datasets from citizen science projects have the advantage of offering low-cost information on long-term temporal and large spatial extents from many different individuals belonging to several populations (e.g., [23]). Prominent electronic citizen science data bases in terms of number of users include eBird [24] and Trektellen [25]. These two biodiversity-related citizen science projects gather records of birds provided by professional and amateur ornithologists around the world. Although eBird is a common data source used by scientist all over the world, Trektellen datasets, specifically addressing migratory bird counts, have remained unexploited by the scientific community to the best of our knowledge.

The Balearic shearwater (*Puffinus mauretanicus*) only breeds from February to June in the Balearic Islands (Spain) and it is considered as critically endangered [26]. Balearic shearwaters spend about one quarter of the year on migration [13]. To date, the existing scientific knowledge on the Balearic shearwater migration is based on two studies using geolocation archival tags conducted on colonies located in the same archipielago: (i) 26 individuals from the large breeding colony in Mallorca (ca. 200 breeding pairs), tracked between 2010–2011 [13], data (ii) on 16 individuals breeding in Eivissa Island (with 310 pairs estimated) monitored during 2011–2012 [15]. Additional surveys using ship transects along the coast of the western Iberian Peninsula from 2004 to 2009 [27], and land- and boat-based surveys in 2007–2010 [14] have also provided partial information on the at-sea distribution of this species during the non-breeding season (July-January). Yet, these studies showed that areas used by shearwaters may change from year to year in relation to interannual variability in the environmental conditions [13, 14], as well as among different breeding colonies [15]. Moreover, with an estimated global population of over 25,000 individuals, according to counts at sea [28], unused areas not detected in these studies may be relevant for other Balearic shearwater populations [29].

During migration, Balearic shearwaters occur in relatively shallow, coastal waters along the shoreline, allowing their observation from land-based sites [28, 30]. Therefore, citizen science projects recording Balearic shearwaters from the coast may provide useful records for the monitoring of this species along its migratory route, completing the partial picture offered by geolocators [13, 15] and vessel surveys [14, 27]. Together with citizen science projects, there is an increasing availability of large-scale environmental data (e.g., satellite imagery), which can be used to predict species distributions over large areas.

Modelling techniques based on these datasets can be used to predict the distribution of the Balearic shearwaters along the coast to identify the most likely marine habitats used during migration, both over time and across space [31]. In contrast to models predicting animal occurrence (such as Species Distribution Models -SDMs-; [32]) studies trying to model estimates of relative abundance have been much less common (but see [27]), even when data are acquired through systematic surveys. Moreover, approaches modelling abundance given only presence have received little attention in Ecological Modelling [33].

We undertook a multi-year study of the spatio-temporal patterns in Balearic shearwater abundance (specifically abundance given only presence) during migration using data gathered within eBird [24] and Trektellen [25]. Particularly, we aimed to: (1) determine general regional environmental predictors driving spatial use during migration, (2) assess whether these variables differ between pre-breeding (September to December) and post-breeding migration and moult (May-August), and (3) determine possible trends in the migratory spatial patterns by identifying latitudinal changes along the migration route in relation to environmental variables.

## Materials and methods

### Study species

The Baleraric shearwater (*Puffinus mauretanicus*) is included as 'Critically Endangered' on the IUCN Red List [26] and it is also considered as threatened bird for the European Union (i.e., rare or vulnerable bird species as listed in Annex I of the E.U. Bird Directive). Most of the population of Balearic shearwater leaves the Mediterranean each year after breeding [13] and mostly remains in the Atlantic during the non-breeding season [34]. At sea, it usually occurs in productive shelf areas related to oceanographic frontal systems [15, 27, 35]. During migration, Balearic shearwaters tend to fly very close to the shoreline [28], with an average off-shore distance of about 1,190 m at the Strait of Gibraltar [36]. Fluctuations in Balearic shearwater

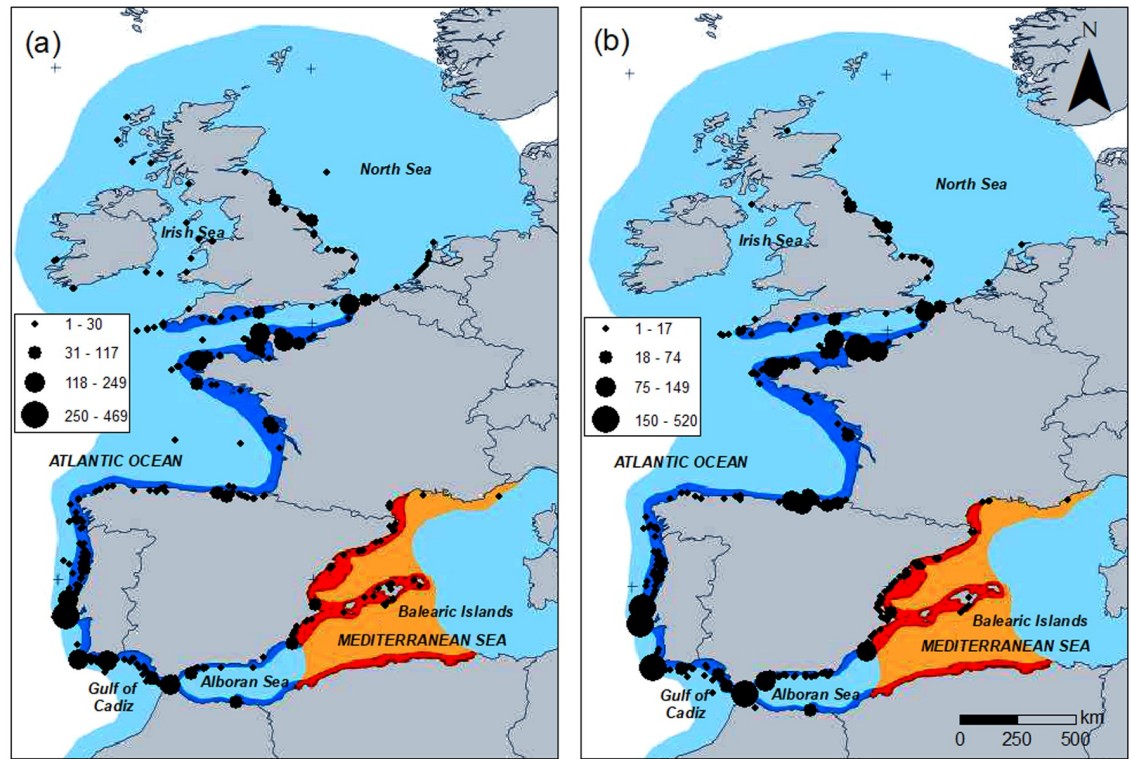

**Fig 1. Balearic shearwater observations.** Spatial distribution of observations of Balearic shearwaters considered in the analysis. a) pre-breeding migration (n = 7,492); b) post-breeding migration (n = 4,690). Period 2005–2017. Symbols are proportional to the total number of sightings in each spatial location. Orange: breeding range, blue: non-breeding distribution; areas darker in colour: common presence, areas lighter in colour: scarce presence; from [34].

migration, both seasonally and inter-annually, seem to be related to changes in food resources [14, 37]. Balearic shearwater's diet includes small pelagic but also demersal fish, frequently obtained from trawling discards. The species can eventually feed on plankton and macrozoo-plankton, specifically krill [38, 39].

The extent of the study area (Fig 1) covers the whole distribution range of the Balearic shearwater throughout the year [26].

## Model variables

**Abundance of shearwaters.**   Daily abundance of the species was recorded from 1964 to 2018, both as opportunistic sighting records and within systematic effort-based surveys (i.e., with an standardized duration of the sampling effort) obtained from the online databases Trektellen [25] and eBird [40]. Only records attributed to Balearic shearwaters were analysed. Balearic shearwater generates much attention both from amateur and expert ornithologists thus, it is usually well-known species and correctly identified when sighted. Due to the opportunistic nature of some of these data, we needed powerful modelling techniques to obtain robust results (see below). Although it was possible to differentiate opportunistic and systematic surveys within the dataset, we opted for keeping all records for our analysis in order to test the robustness of the modelling approach in case our methods will be extended to other species for which this information is not available. Since some records contained in Trektellen database may be also included in eBird, based on date and spatial location, we avoid duplicated observations between databases. From Trektellen database, according to the migratory range

of the study species, we specifically obtained observations from 123 sighting points in France, Portugal, Spain and UK, from 2005 to 2018, counting 7,619 different observations and totaling 378,112 birds. Regarding eBird records, there are not fixed sighting points, although we obtained data on 258,730 birds observed in 8,244 different records from 1964 to 2018. However, to ensure the maximum seasonal representation of the dataset to be analysed (see S1 Fig in S1 File), from this total sample of birds, we only modelled data collected during the migration period from 2005 to 2017 (see Results). We defined migratory periods based on the known phenology of the Balearic shearwater. Specifically, we considered 'post-breeding migration', the northward migration of birds leaving the Mediterranean and molt, between May 1st—August 30th, and 'pre-breeding migration', corresponding to the southward migration of birds from the Atlantic and returning to the breeding areas in the Mediterranean, from September to December [41, 42]. Birds at their breeding grounds in the Balearic Islands (i.e., birds observed at land) were removed from the data sample.

**Response variable and environmental predictors.** Our response variable was Balearic shearwater abundance (i.e., only abundance given presence thus, absence of abundance -i.e., zeros- was not considered in the analysis) [33], expressed as the number of birds sighted on a given date at a given latitude / longitude, during migration. Shearwater abundance was modelled using 15 environmental predictors (Table 1, see S1 File) that have been previously described to be related with the spatial distribution at sea of this and other seabird species [4, 14, 27, 39–44].

In addition to the previous environmental variables, as discrete variables predicting shearwater abundance we also considered spatial coordinates (longitude and latitude), as well as date (i.e., julian date) and year of the observation for explicitly modelling seasonal and interannual variation in shearwater abundance, respectively. Fluctuations in seabird abundance during migration are closely related to changes in food resources (see Supplementary Methods in S1 File). However, photoperiodic cues and/or endogenous rhythms may also modulate seabird breeding and migration periods [43]. Therefore, apart from food availability, migration

**Table 1. Description of environmental predictors.**

| Name | Description | unit | Source | Period |
|------|-------------|------|--------|--------|
| **batim** | bathymetry | meters | EMODnet (*European Marine Observation and Data Network*) | 2012 |
| **fish** | fishing intensity | number of vessels | JRC Data Catalogue; "Automatic Identification System" (AIS) | 2014–2015 |
| **tmmean** | mean temperature | ˚ K | NCEP/NCAR Reanalysis dataset (NOAA ESRL Physical Sciences Division) | 1964–2018 |
| **tmstd** | standard deviation of mean temperature | ˚ K | NCEP/NCAR Reanalysis dataset (NOAA ESRL Physical Sciences Division) | 1964–2018 |
| **uwmean** | mean wind speed (u-wind: east-west direction) | m/s | NCEP/NCAR Reanalysis dataset (NOAA ESRL Physical Sciences Division) | 1964–2018 |
| **uwstd** | standard deviation of mean wind speed (u-wind: east-west direction) | m/s | NCEP/NCAR Reanalysis dataset (NOAA ESRL Physical Sciences Division) | 1964–2018 |
| **vwmean** | mean wind speed (v-wind: north-south direction) | m/s | NCEP/NCAR Reanalysis dataset (NOAA ESRL Physical Sciences Division) | 1964–2018 |
| **vwstd** | standard deviation of mean wind speed (v-wind: north-south direction) | m/s | NCEP/NCAR Reanalysis dataset (NOAA ESRL Physical Sciences Division) | 1964–2018 |
| **clorof** | chlorophyll concentration | mg m-3 | JRC Data Catalogue; MERIS | 2004–2017 |
| **NAO** | NAO index | - | Climate Prediction Center (US National Weather Service, NOAA) | 1950–2018 |
| **moon** | daily fraction of the moon illuminated at midnight | % | U.S Naval Observatory & Astronomical Applications Department | 2005–2018 |

decisions in Balearic shearwaters might be partially dictated by daylength and/or an internal rhythm making the bird instinctively moving into the west-north, as long as the post-breeding season progresses, and then into the south-east during the pre-breeding period. In this sense, date, longitude and latitude variables allow us to include in the models the endogenous rhythm of the bird. In addition, longitude and latitude can be indirect proxies of the effects that variable wintering sites, length of the route and en-route environmental conditions may pose to different migrant shearwaters. Finally, due to potential differences in the rates of change of the environmental predictors across space, interactions between predictors and "latitude" and longitude, and between "year" and "latitude", may allow to quantify both the spatial and temporal heterogeneity in the migratory responses.

## Statistical analysis

To avoid collinearity, from the total set of environmental predictors the correlation between pairs of variables (i.e., Pearson correlation coefficient) was assessed (see Supplementary results in the S1 File).

We applied machine-learning techniques to predict shearwater abundance, specifically Random Forest regression models. This choice was based on a previous assessment of eight different modelling techniques carried out on the same dataset [44]. Specifically, Generalized Additive Models (GAM), Classification and Regression Trees (CART), Bootstrap Aggregation (bagged CART), Extreme Gradient Boosting, Stochastic Gradient Boosting, K Nearest Neighbours (KNN), Support Vector Machine (SVM) and Multilayer perceptron Neural Network (MLP). Comparisons among modelling techniques were based on the Root Mean Square Error (RMSE; the average difference between the observed known values of the outcome and the predicted value by the model), and on the amount of variation explained ($R^2$), measured as the mean-squared error, divided by the variance of the original observations [45]):

$$R^2 = 1 - \frac{\sum_i (y_i - \hat{y}_i)^2}{\sum_i (y_i - \bar{y}_i)^2}$$

RMSE and $R^2$ values were derived from a cross-validation procedure after randomly splitting the abundance data into training and test data (setting aside 20% of the data for testing the models). Although the differences between models were not always statistically significant at a Bonferroni corrected p-level, the assessment of all these modelling techniques showed that Random Forest performed better in terms of RMSE [46], $R^2$ and in terms of the correlation between observed and predicted abundance. Therefore, we used packages caret [47] and randomForest [48] in R [49] to build Random Forest models in the present study. We built separated models for pre-breeding and post-breeding migration. As usual in count data, abundance of shearwaters followed a Poisson distribution [50]. Although there is a lack of assumptions in the distribution of the data in Random Forest models, when we use an approach based on decision trees such as this where data partitioning is applied, we can obtain better results if we model a dependent variable homogenously distributed, because the model dispersion increases as long as the variable increases. Therefore, prior to build our models, we log-transformed (i.e., natural log) the abundance data. This transformation increased the variance explained by the models (from 38% to 52.55% during pre-breeding, and from 30.79% to 52.89% in the case of the post-breeding best-fit models; see Results). All the random forest trees were built using the total set of predictors described in Table 1.

Any random forest model apply bagging (i.e., bootstrap aggregating) to sub-sample the data that are used for training, thus each new tree is fit from a bootstrap sample of the training

observations $z_i = (x_i, y_i)$. We assessed the performance of our models based on the accuracy of predictions derived from the Out- of-bag (OOB) error [46]. The OOB error is the average error for each $z_i$ calculated using predictions from the trees that do not contain $z_i$ in their respective bootstrap sample [51]. This allows the model to be fit and validated whilst being trained, thus no additional cross-validation was required. On the other hand, Random Forest has hyperparameters that must be tuned to avoid overfitting. To determine the parameter values offering the best fit, we specified a set of tuning values to be tested during the calibration of the models. Specifically, we applied a grid search method, thus we evaluated the model over different combinations of parameters included in the grid (values ranging between 1–15, at one-unit intervals). When bagging the number of samples and hence the number of trees is also a parameter to be selected. The optimal number of trees in the models was determined from a range between 100 and 500 trees in such a way that the algorithm increases the number of trees on run after run until the accuracy does not significantly improve. To identify the model with the optimal parameter combination and number of trees (i.e., offering the best fit) we compared the RMSE values of the models [46]. As an additional assessment of the models, we quantified the amount of variation explained ($R^2$), as a measure of how well out-of-bag predictions explained the target variance. The differences in RMSE and $R^2$ between pre-breeding and post-breeding models were measured as the lagged and iterated differences over the model resamples in the bagging procedure.

To evaluate how well the models were able to predict the test set outcomes, we randomly split the abundance data into training and test data subsets by setting aside 20% of the data for testing the models [52]. Additional models were then calibrated on the training data and then evaluated on the test data. Specifically, we quantified to what extent the abundance in test data agreed with the model predicted abundance by means of the Pearson correlation coefficient.

Relative importance of the variables used for predicting shearwater abundance was assessed by means of the number of trees where the variable was included and the minimal depth of the variable in the tree (package randomForestExplainer in R). Minimal depth allows to determine variable importance by the position of the variables in the decision trees in such a way that the importance of the variable is based on the decision tree structure. In this way, variables that tend to split close to the root node of a tree should have more importance in prediction [53]. Relative importance, as well as total variance explained, were derived from models built with the total dataset (both training and testing).

As an example of the potential marine areas that can be identified from the models, we predicted shearwater abundance across the study area on the fifteenth of the month, from May to December, based on the environmental conditions during 2017.

## Results

An exploratory analysis of the data (see S1 Fig in S1 File) showed that observations of Balearic shearwaters contained in the databases preceding the year 2005 and collected during 2018 were insufficient to model daily patterns during the migration periods. After the removal of data recorded before 2005 and after 2017, 7,492 and 4,690 observations (i.e., rows in the data matrix) remained for the pre-breeding and post-breeding migration periods, with a total of 233,863 birds during pre-breeding and 151,782 during post-breeding, respectively. These observations covered the entire non-breeding range of the species (Fig 1).

After inspection of collinearity (i.e., significant Pearson correlation coefficient between pairs of variables), we did not find highly correlated variables (r<0.3 for the total set of environmental predictors in Table 1; see S2 Fig in S1 File), thus all the predictors considered in the initial set were kept for model selection.

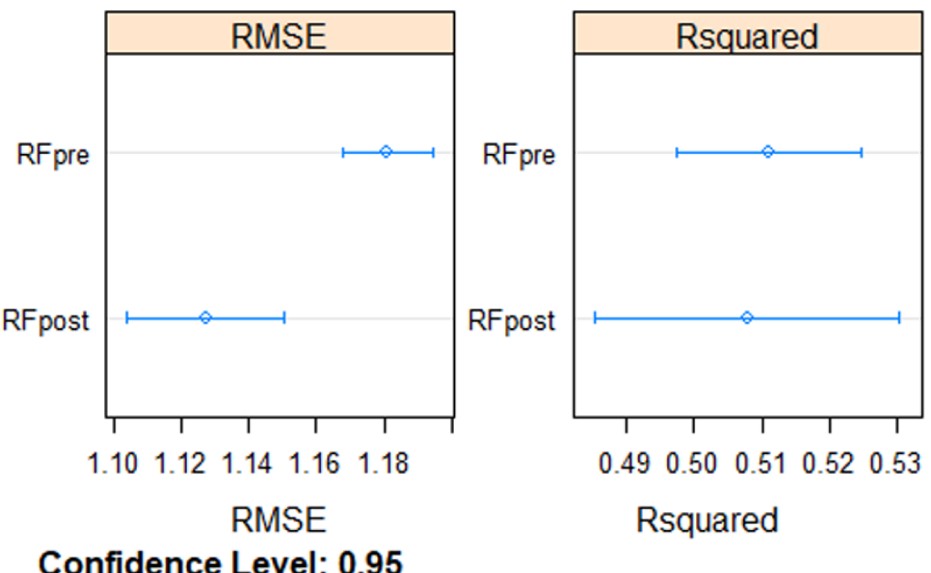

**Fig 2. Model comparison.** Comparison of resamples (bagging) between pre- (RF pre) and post-breeding (RF post) models. *RMSE*: Root Mean Squared Error; *Rsquared*: % variance explained. Mean and Confidence Interval (95%).

According to the results derived from the bagging procedure, there was substantial variation among models built with different training data subsamples (measured as 95% confidence interval) in terms of RMSE and $R^2$, particularly during the post-breeding migration (Fig 2). The estimates of the differences between pre-breeding and post-breeding models (i.e., lagged and iterated differences over the model resamples), even though small, show that the accuracy (RMSE) of the pre-breeding model was 3% significantly larger compared with the post-breeding model (Bonferroni's p-value adjustment < 0.01; p-value for H0: difference = 0). Contrastingly, differences in the variance explained ($R^2$) between the pre-breeding and the post-breeding migration models were not statistically significant (Bonferroni's p-value adjustment = 0.7314). Predictive ability was substantially high during the post-breeding period, when the correlation between observed and predicted abundance was 0.71, as well as in the pre-breeding model, where the correspondence between observed and predicted values was 0.68. Models built with the total dataset for both pre-breeding and post-breeding migration explained around 53% of the variation in shearwater abundance (computed from bootstrap resampling with 25 repetitions).

Variable importance differed between migration periods (Fig 3), with a larger importance of chlorophyll concentration during the post-breeding migration. Bathymetry showed a moderate contribution to the abundance of shearwaters during both migration movements. If there was no interannual variability in the spatial patterns of shearwater abundance, then the "year" predictor should have a minor importance in the models. However, interannual variability (i.e., effect of "year") was relevant during pre-breeding and post-breeding migration. In relation to weather predictors, temperature and wind were variables significantly affecting the abundance of shearwaters during both migration periods. In contrast, lunar cycle and NAO index hardly contributed to the abundance patterns described in the models.

According to the predicted abundance of shearwaters for different values of year and latitude, the number of migrating shearwaters appeared to be larger at higher latitudes in recent years, although there was no clear trend over the study period (Fig 4).

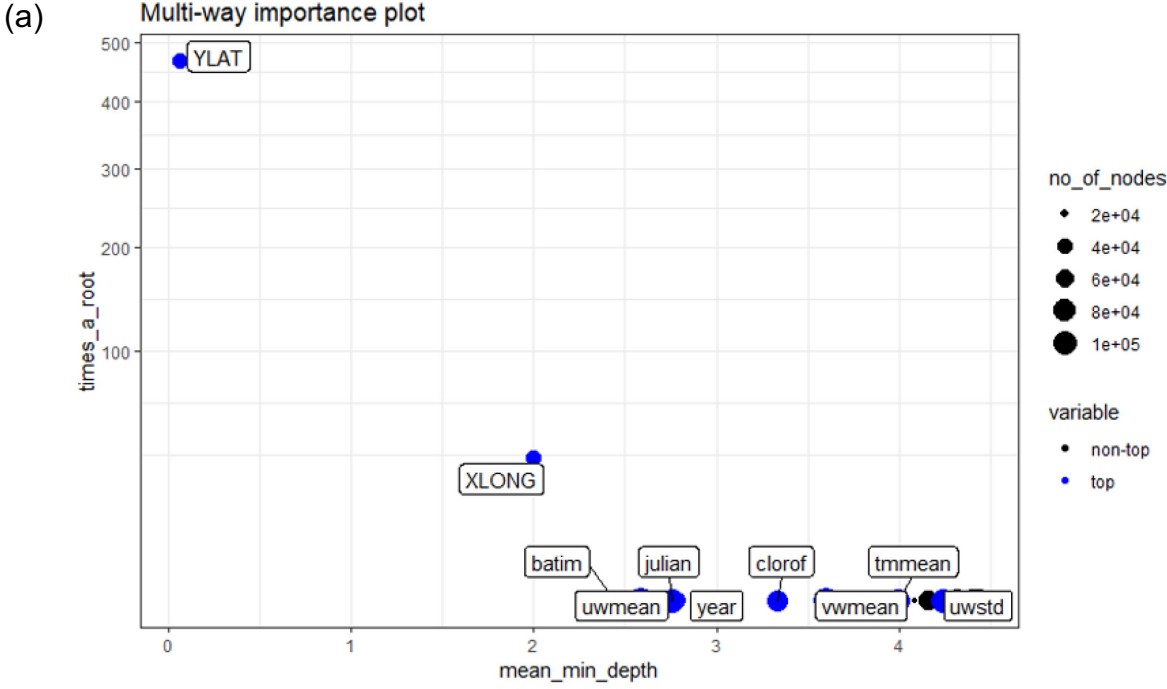

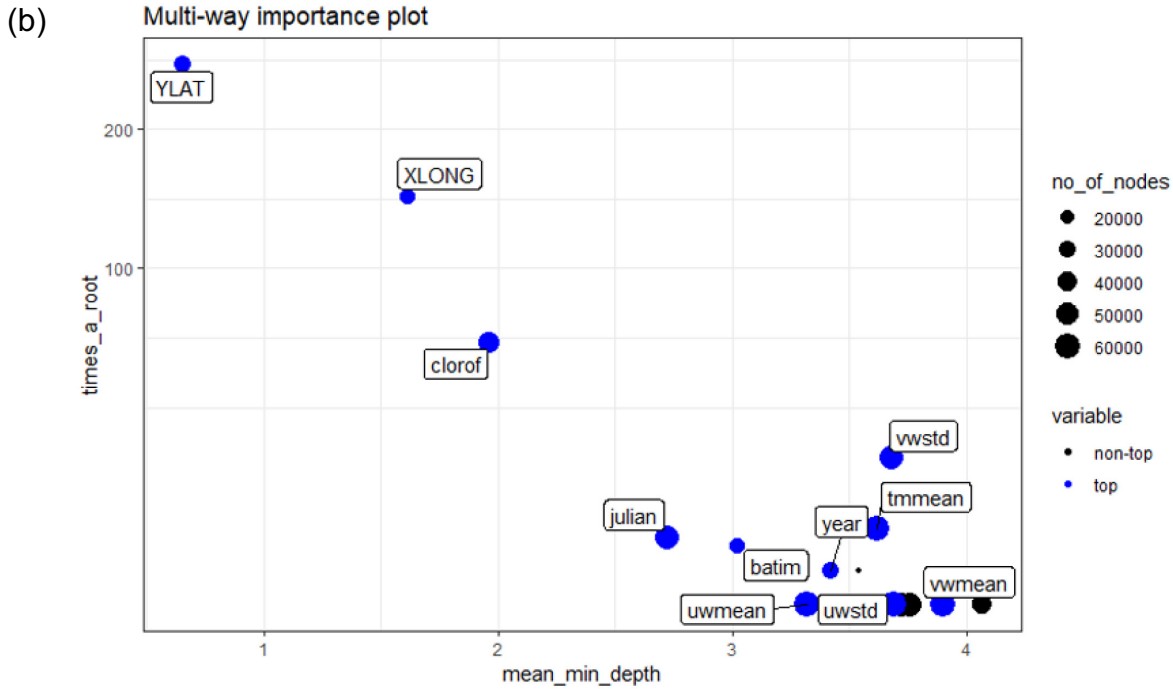

**Fig 3. Relative importance of the variables.** Relative variable importance (in terms of number of trees and minimum depth) in the random forest models predicting Balearic shearwater abundance. a) pre-breeding migration; b) post-breeding migration. *times_a_root*: total number of trees in which the variable is used for splitting the root node. *mean_minimal_depth*: mean minimal depth. Ten top variables are highlighted in blue. The size of points reflects the number of nodes split on the variable.

(a)

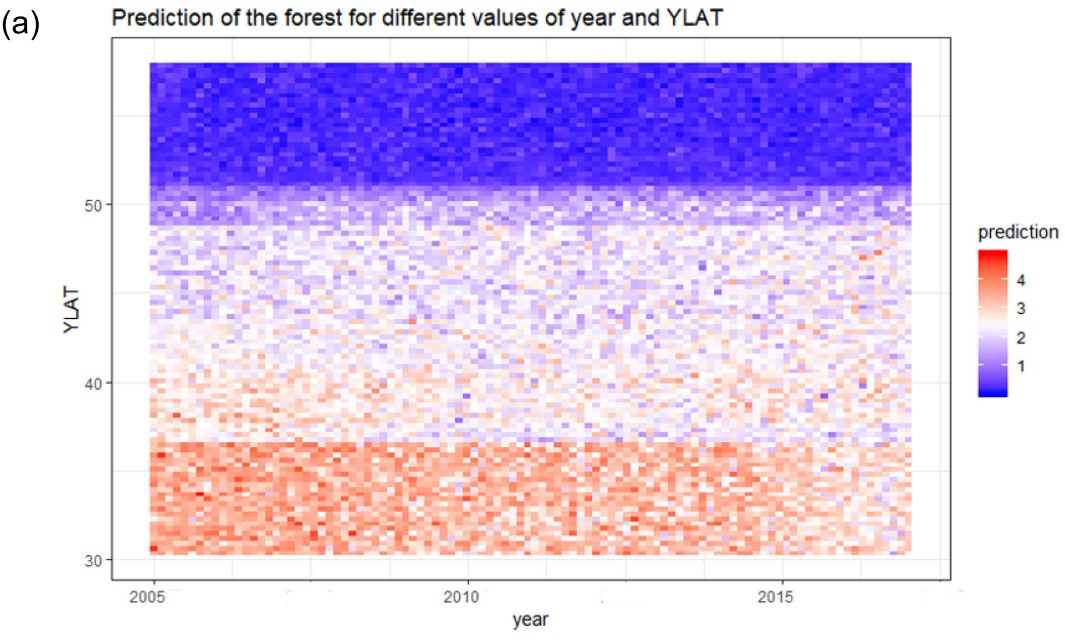

(b)

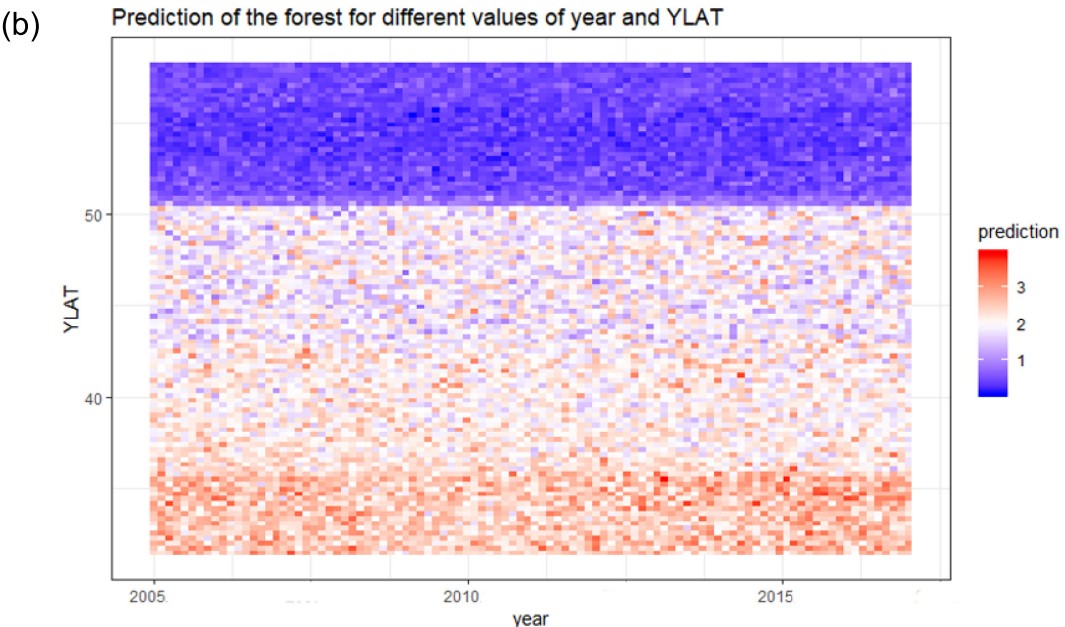

**Fig 4. Interactions between variables.** Predicted abundance of shearwaters (pedictions in the natural logarithmic scale) for different values of year and latitude. (a) pre-breeding; (b) post-breeding migration models.

Spatial predictions derived from pre- and post-breeding models (Fig 5) showed how abundance of shearwaters decreases at sea locations in the Mediterranean from May to August and then it gradually increases again from September to December. Overall, according to the models, Balearic Shearwaters use slightly different regions during their northward (May–August, post-breeding) and southward migration (September–December, pre-breeding).

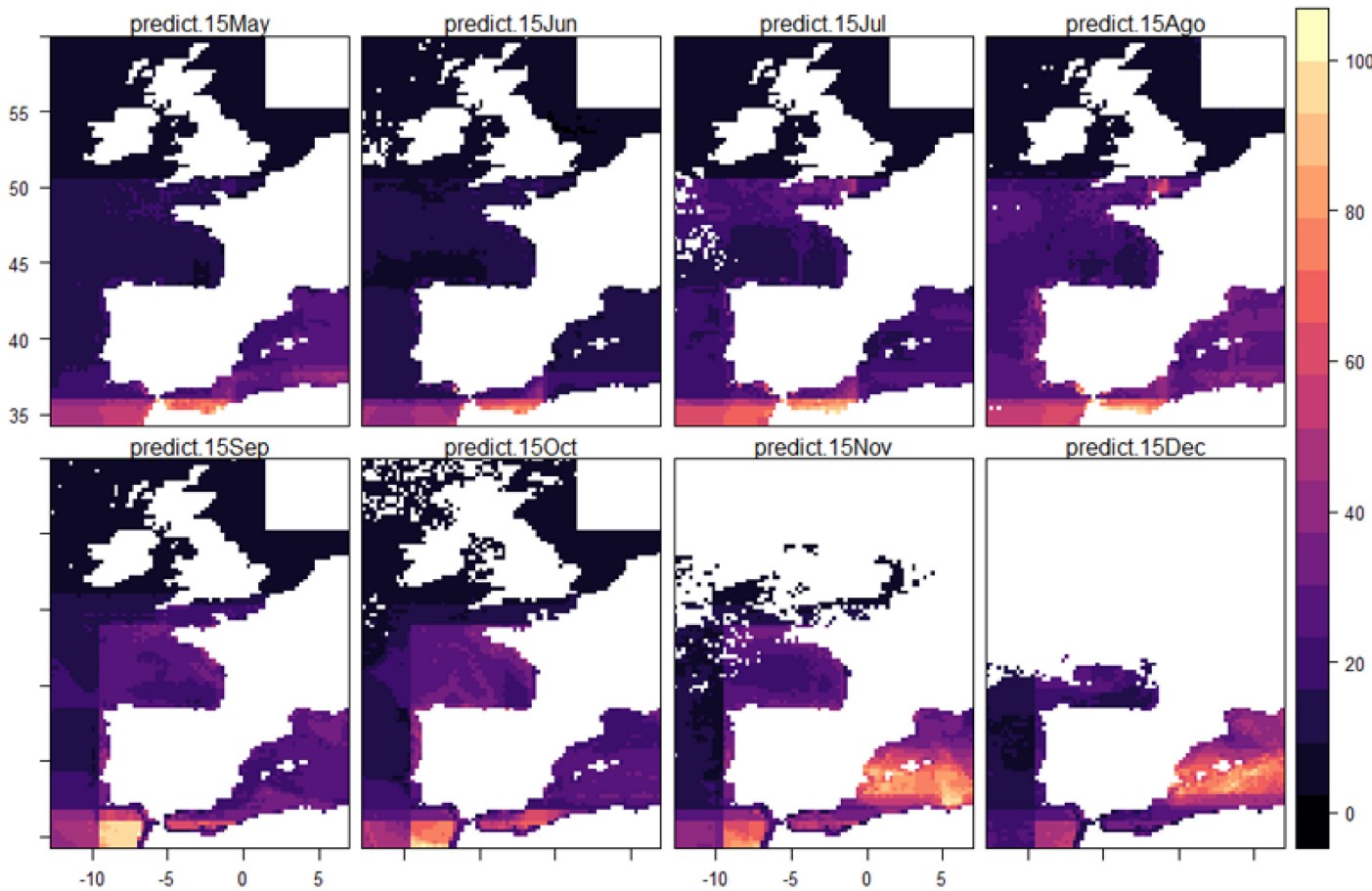

**Fig 5. Predicted abundance of Balearic shearwater across its distribution range during migration.** Back-transformed predictions. Colour gradient indicates the percentage of the maximum predicted abundance across the study area on the fifteenth of the month, from May to December (May-August: post-breeding; September-December: pre-breeding), from environmental conditions (see Table 1) occurred in year 2017.

## Discussion

Here we undertook the first long-term study on the spatio-temporal patterns of migrating Balearic shearwaters from data gathered within the framework of citizen science projects. Our results showed that data obtained in initiatives such as Trektellen and eBird, together with large-scale open datasets (NCEP/NCAR, AIS, NOAA, among others), can be successfully applied to describe the migratory distribution and abundance of seabird species accurately. Although these large volumes of data maybe hard to analyze and interpret, and relevant interpretations may be limited in scope, accurate information on the distribution of species during the entire annual cycle can be derived from these datasets, even for populations of long-distance migratory bird species [24].

Presence-only SDMs (Species Distribution Models) are based on the relationship between species presence and environmental conditions, predicting the environmental suitability for a particular species, but not its actual distribution. This leads to weak relationships between predicted presence and the relative abundance of the species [54]. In contrast the above-mentioned approaches, models based on abundance given only presence have received little attention in species distribution research [33]. Previous research found Balearic shearwater

abundance to be extremely difficult to predict from and abundance-absence model, and it failed to provide reliable predictions on the spatial distribution of shearwater numbers [27]. In contrast to this earlier modelling attempt based on ship transect data, our models based on abundance given only presence of shearwaters provided strongly correlated predictions with the observed abundances and explained a significant proportion of the variance existing in the numbers of migrating shearwaters.

In any research, the value of the derived knowledge directly relies on the quality of the data used. Direct observations of birds collected by volunteers are a cost-effective source of data that have been previously applied to develop spatial distribution maps of seabirds [55]. Nevertheless, observations of migratory movements collected from coastal land-based observatories have been claimed to be fragmentary and biased thus, they have been frequently considered to be deficient in determining migratory patterns [30]. In recent years, however, the new Big Data approach has led to an increased interest in gathering massive quantities of datasets frequently shared over the Internet. As other Big Data, observations derived from citizen initiatives frequently exhibit noise affecting their quality [56]. To deal with such a large amount of data and its characteristic noise, machine learning techniques (i.e., the application of computational methods underlying experience-based decision making; [57], as those applied in our study, offer an efficient tool, bringing new opportunities to take advantage of these massive data [58]. In contrast to alternative approaches such as the use of tracking data [12, 13, 29] or vessel surveys [14, 27], these massive datasets provide with low-cost information on long-term temporal and large spatial extents. This allows to identify spatio-temporal patterns from many different individuals belonging to several populations (e.g., [23]), as well as to quantify their seasonal and interannual variation in the long-term. Contrastingly, citizen science data have one main drawback which is low data quality, due to low accuracy and precision, insufficient sample sizes as well as insufficient temporal and spatial representation [22]. However, the members of a community of citizen scientist have frequently training and expertise, thus they can be considered as "expert amateurs". Thanks to the expertise and high level of interest in the topic of this citizen scientists makes that the identification of the study objects is similar between experts and citizen scientists. This is particularly the case for eBird and Trektellen participants, who must be familiar with, or at least interested on bird identification and, frequently have as good or even better identification skills as professional ornithologists [59]. In addition, thanks to the large number of participants in these citizen science projects, our results show the high level of temporal and spatial representation of these datasets.

Our predictions should be taken rather as an illustration than as an exact calculation, since they are based on a single snapshot on the fifteenth of the month in a particular year. However, they support the validity of our models results, showing that the spatial predictions obtained are consistent with previous findings on important marine areas for shearwaters identified from tracking data and vessel surveys [13, 14, 27]. Although we are analysing no behavioural data, to some extent, we can infer feeding and transiting areas from variable interactions (see S4–S6 Figs in S1 File). For instance, locations with high abundance level, tailwinds, and low chlorophyll concentration can be interpreted as locations with high flux of birds, in contrast to locations with high-moderate abundance, no winds or moderate winds and high chlorophyll concentration levels. Feeding or transiting, the predictions obtained supported that Alboran Sea is an important area for migrating shearwaters [60], particularly during post-breeding migration. Similarly, Gulf of Cadiz also showed high predicted abundance of shearwaters [61], especially during the pre-breeding movement. Our results also suggested that Atlantic coast of Portugal and France are migratory, stopover and/or moulting sites for Balearic shearwaters both during pre and post-breeding migration. Similarly, the post-breeding model showed that Western English Channel is an important area for Balearic shearwaters mainly during August,

whereas high abundance of shearwaters is predicted along the Algerian/Tunisian coastline in November. Furthermore, model predictions support previous research, which shows that a variable fraction of the total Balearic shearwater population, mainly adult birds, appeared to remain in the Mediterranean all year round [62].

But the major accomplishment of our modelling approach is the high temporal resolution that we achieved with our models and the possibility of deriving daily spatial predictions that take into consideration the spatio-temporal heterogeneity in migration patterns for specific time frames. In this sense, most of the studies predicting animal spatial patterns have been focused on the stationary distribution in more or less temporally invariable environments (but see [63, 64]). In contrast, marine environment is highly dynamic and its conditions in a particular location are ever changing [65]. New available global data from satellite imagery are frequently provided at high spatial and temporal resolutions, offering an excellent opportunity to model highly mobile migratory species by informing on the heterogeneity at different spatio-temporal scales affecting the en-route habitat use [17]. This high-resolution information allows modelling daily or even hourly conditions at specific locations, such as in our study. We show that these data, together with observations of birds collected in citizen science projects, can be used to identify conservation concerns and targets at regional levels related to environmental changes such as global warming or fisheries, among others, that are difficult to track using traditional approaches.

## Differences between pre- and post-breeding migration

In addition, we show that predictive models based on citizen science data not only provide accurate predictions but they contribute to a better understanding of the factors modulating the migratory periods and to identify the seasonal and interannual variability existing in the at sea locations during migration. Among others, fluctuations in migration abundance are usually related to changes in food resources [37]. Differences in chlorophyll-a concentration and bathymetry of the at sea distribution of Balearic shearwaters between the two migration movements suggest that birds leaving the Mediterranean target shallower waters where productivity is higher, whereas shearwaters returning to the breeding grounds appear to be less conditioned on food availability. In this way, the distribution of the shearwater abundance during the pre-breeding period depends more on static predictors such as spatial location (i.e., longitude and latitude) and bathymetry, whereas during the post-breeding it is more dependent on higher spatially and temporally variable predictors such as food availability (i.e., chlorophyll). Most likely, this is because, after breeding, migrating birds need to replenish energy reserves, thus they will select key stopover locations along the trip where maximize their refueling opportunities [66]. This stronger link between shearwater abundance and food availability likely lead to the relatively larger variability (in terms of statistical significance) observed in the spatio-temporal patterns of shearwaters during post-breeding compared to the pre-breeding migration.

Trends in abundance related to changes in fisheries and/or in climate affecting the Balearic shearwater distribution can be inspected from a range of latitude and year values over the study period. Although there is no a straightforward pattern over the study period, it seems that the abundance of the Balearic shearwater has increased in northern latitudes in recent years. These results are consistent with the northward shift in the spatial range of Balearic shearwaters along northwest European coasts that has been found in previous studies [37, 67, 68], apparently not directly related to climate change but meditated by recent changes in prey fish and discard availability [14].

## Predictive models for delimiting marine protected areas

Understanding the spatial distribution of seabirds at sea is crucial for the protection of these species and their habitats [1]. However, whereas the terrestrial breeding areas where seabirds occur are usually well known, a gap still exists in the identification of protected areas at sea (e.g., Natura 2000 network; [69]. Identification of marine protected areas for seabirds and other highly mobile marine species is a difficult task [70] because their movements at sea and the spatio-temporal variability associated remain largely unknown [29]. In this sense, a number of more or less successful approaches have been developed to identify marine Important Bird and biodiversity Areas (IBAs), which are important foraging, migrating or wintering sites for seabirds [71]. Our model predictions can be useful for delimiting important areas for the conservation of the Balearic shearwater. Specifically, mapping model average and/or accumulate predictions over seasons and/or particular interannual time frames can provide useful information on migratory routes and stopover sites, as well as on the seasonal and interannual variability in such factors and patterns.

## Caveats regarding data availability and predictive ability

According to our results, the predictive ability of the models describing Balearic shearwater abundance patterns (particularly during post- breeding migration when according to the RMSE values the variability in accuracy of the predicted abundance is larger) is subjected to the dataset used in the model calibration. Balearic shearwater is a rare but a charismatic species which generates high interest among birdwatchers. This fact, and its mostly coastal spatial distribution, facilitate an abundant and correctly identified record of this species from land-based sites throughout its annual cycle and across its non-breeding range. In contrast, models built for other species less well-known and/or more difficult to observe could not offer as good results as those here shown for Balearic shearwaters. However, according to our results and considering the increasing amount of data collected by volunteers in the framework of citizen science initiatives all over the world, these datasets should be taken into consideration when studying highly mobile animals.

## Conclusions

We show that data gathered in citizen science initiatives together with recently available high-resolution satellite imagery may offer a powerful and cost-effective tool for the long-term spatial monitoring of the migratory patterns in sensitive marine species. Due to cost and logistic constraints, long-term monitoring of seabirds is scarce [72]. In addition, seabird behavior at sea is complex, with high variability in the locations mainly driven by spatial and temporal changes in food availability affecting the size and the shape of the areas used. As an alternative to more traditional approaches [73], modelling techniques can provide a comprehensive picture, both spatially and temporally, of the migratory patterns at the population level that can be complementary to the detailed information at individual level obtained from tracking data. Similar machine learning techniques as those applied in this study, may contribute to extract information from other existing and future datasets collected by volunteers and inform marine spatial planning at regional spatial scales for multiple species. Modelling approaches can also become key tools to detect the impacts of climate and other global changes in the at sea distribution of this and other marine species within the range of the training data. In this sense, other seabird and sensitive vertebrate marine taxa of conservation concern such as sea turtles [74] and marine mammals [75] could also benefit from our approach.

## Supporting information

**S1 File.**
(DOCX)

## Acknowledgments

This study is part of a research project ("Environmental factors determining the interannual variation in the migration of Balearic and Scopoli's shearwaters in the Mediterranean", 2018–2019) which form part of the Annual Programme of Grants of the *Instituto de Estudios Ceutíes* (IEC, Autonomous City of Ceuta, Spain), years 2018–2019. We would like to thank the editor and three anonymous referees for providing us with comments and suggestions which really helped to improve the manuscript.

Regarding the data used in this study, Trektellen is a public database of migration / sea-watch counts and ringing results (https://www.trektellen.nl/). In addition, terms of use of eBird database [40] have been cleared (https://www.ala.org.au/wp-content/uploads/2015/02/Terms_of_Use.v3.pdf). All the environmental predictors used in the present study are freely available from the websites of the different data sources indicated in Table 1 as well as from Trektellen website. Data sourced by eBird can be obtained after registration and request at https://ebird.org/.

## Author Contributions

**Conceptualization:** Beatriz Martín, Alejandro Onrubia.

**Data curation:** Beatriz Martín.

**Formal analysis:** Beatriz Martín, Julio González-Arias, Juan A. Vicente-Vírseda.

**Funding acquisition:** Beatriz Martín.

**Investigation:** Beatriz Martín.

**Methodology:** Beatriz Martín.

**Project administration:** Beatriz Martín.

**Resources:** Beatriz Martín.

**Software:** Beatriz Martín.

**Supervision:** Beatriz Martín, Julio González-Arias, Juan A. Vicente-Vírseda.

**Validation:** Beatriz Martín, Alejandro Onrubia.

**Visualization:** Beatriz Martín.

**Writing – original draft:** Beatriz Martín.

**Writing – review & editing:** Beatriz Martín, Alejandro Onrubia, Julio González-Arias, Juan A. Vicente-Vírseda.

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
