## [Decision Letter · Decision Letter 0]

8 Jan 2020

PONE-D-19-29580

Citizen science and machine learning for predicting spatio-temporal patterns in seabird migration

PLOS ONE

Dear Dr Martin,

Thank you for submitting your manuscript to PLOS ONE. After careful consideration, we feel that it has merit but does not fully meet PLOS ONE’s publication criteria as it currently stands. Therefore, we invite you to submit a revised version of the manuscript that addresses the points raised during the review process.

i apologize for the long handling time, but securing reviewers proved very difficult.  Because I was only able to secure one review, I did another review of the ms, and I concluded that it needed substantial revisions before it could be published.  I forward an annotated version with inserted comments, edits, and questions.  In particular, I urge the authors to:

1) revise the introduction to remove redundancies / and focus focus on the main goals.

2) revise the introduction and discussion to ensure the limitations of citizen science data are acknowledged\\. currently, the ms highlights the limitations of tracking and vessel-based surveys, but there are many similar limitations to citizen science data: uncertainty of identification, uncertainty of counts, uneven coverage and lack of systematic effort, and lack of knowledge of the colony provenance of the birds sighted at sea.

3) revise the methods section (and augment with supplementary tables) to provide readers with a sense of how the sighting / fishery / environmental datasets were cleaned / processed. in particular, address the uncertainties associated with the species identifications, and explicitly explain how the fishery data were subsetted by sector / fishery / target.

4) furthermore, please provide evidence that the authors had cleared the terms of use of these publicly available sighting data, and are thus able to publish these datasets.  moreover, explicitly state how many records / birds were included in each dataset, for the pre and post breeding migration. finally, explicitly state whether "zero" counts were included, if standardized surveys (or sightings of other species were reported, but no Balearic Shearwatyer sightings were reported).

5) the methods also need to be strengthened, since the ms refers to several exploratory analyses (daily abundances used to censor the study period and co-linearity of explanatory variables), but no results are provided. I urge the authors to provide supplementary tables with these results, so the readers can evaluate what was done.

6) the results also need to be strengthened, to ensure tests are clearly reported and evaluated. in particular, please use the comments added to the ms to address the comparison of the model R-squared values, and the correlations or modeled / observed abundances. finally, please justify the small number of iterations used (25) and the use of means +-/ 95% CI for some of the estimated parameters, rather than medians (25% - 75%).

7) The discussion and conclusions were clear and concise, but they are difficult to evaluate , given the paucity of details in the methods / results.  Finally, I urge the authors to carefully consider how they identify "key" areas, and to consider that they are only using counts; there are no behavioral data (eg., feeding versus transiting).  Namely, would a high flux area (e.g., Gibraltar Strait) with lower counts be more sensitive to impacts than a foraging area with higher counts but lower flux? 

We would appreciate receiving your revised manuscript by Feb 22 2020 11:59PM. To enhance the reproducibility of your results, we recommend that if applicable you deposit your laboratory protocols in protocols.io, where a protocol can be assigned its own identifier (DOI) such that it can be cited independently in the future. For instructions see: http://journals.plos.org/plosone/s/submission-guidelines#loc-laboratory-protocols

We look forward to receiving your revised manuscript.

Kind regards,

David Hyrenbach, Ph.D.

Academic Editor

PLOS ONE

Journal Requirements:

'This study is part of a research project (“Environmental factors determining the interannual variation in the migration of Balearic and Scopoli’s shearwaters in the Mediterranean”, 2018-2019) which has been partly financed by the Annual Programme of Grants of the Instituto de Estudios Ceutíes (IEC, Autonomous City of Ceuta, Spain), years 2018-2019.'

Reviewers' comments:

Reviewer's Responses to Questions

**Comments to the Author**

1. Is the manuscript technically sound, and do the data support the conclusions?

Reviewer #1: Yes

2. Has the statistical analysis been performed appropriately and rigorously? 

Reviewer #1: Yes

3. Have the authors made all data underlying the findings in their manuscript fully available?

Reviewer #1: Yes

4. Is the manuscript presented in an intelligible fashion and written in standard English?

Reviewer #1: Yes

5. Review Comments to the Author

Reviewer #1: In this manuscript, authors present a novel work to evaluate whether data from a long-term monitoring programme carried out by volunteers, i.e., a citizen science project, can be used to assess bird migratory patterns. They focused on an endangered seabird species endemic to the Mediterranean Sea that migrates to the Atlantic out of the breeding period. Authors relied on supervised machine learning technics to predict bird abundance, building up the models from data gathered at a network of bird census stations. Specifically, they used random forest regression models to evaluate the usefulness of citizen science data in predicting spatial and temporal abundance based on a set of predictors, including many satellite-derived features, reaching an accuracy of about 70%. Authors concluded that combining long-term citizen science with predictive modelling can be a reliable tool to assist long-term monitoring of sensitive species, to identify important areas and/or to detect trends.

The paper is well structured. The study is accurate and provides an interesting approach. As general comments, the major strength of this study relies on the source data used. Trektellen is an online database recording opportunistic seabird counts carried out by volunteers across Europe. Despite its potential, very few papers have made use of this long-term database. As a counterpart, there are also some weaknesses in the work that should be amended for publication. My main concern in this regard is that authors tend to overstate along the manuscript the importance of random forest (RF) to predict spatial and temporal abundance. RF is a really powerful algorithm (I personally really love it) based on an ensemble of decision trees, but its use is not new in the context of species distribution/abundance modelling (e.g. Oppel et al. (2012) Comparison of five modelling techniques to predict the spatial distribution and abundance of seabirds. Biological Conservation, 156, 94-104). In fact, recent approaches for species distribution modelling tend to build up a final output based on an ensemble of several algorithms (a.k.a. Ensemble Ecological Niche Models), including RF among them. See the “ssdm” package in R for details (Schmitt et al. (2017) ssdm: An r package to predict distribution of species richness and composition based on stacked species distribution models. Methods in Ecology and Evolution, 8(12), 1795-1803; and also, e.g., Pereira et al. (2018) Using a multi-model ensemble forecasting approach to identify key marine protected areas for seabirds in the Portuguese coast. Ocean & Coastal Management, 153, 98-107.). Apart from that, a major weakness of RF, just as in decision trees, is that they cannot extrapolate outside the range of the training data, and therefore great care must be taken when these algorithms are used to predict trends in the long-term, as those expected in the context of climate change. For these reasons, I think authors should attenuate the importance of using this machine learning algorithm in the paper, beginning with removing it from the title.

Below I will refer to specific section and lines to provide comment:

Title

As I have already commented, I would recommend changing the title, omitting reference to machine learning on it.

Abstract:

There is a lack of introduction. Please include at least one or two sentences about the general framework before stating what you did address.

Introduction:

In general, I think the introduction needs a bit of work. Some sentences are not explained well enough.

The first paragraph is not very cohesive.

L49: Should be rephrased. Although I understand what authors mean, these concepts could be explained in a clearer way.

L52-53: I would say seabirds are threatened because they are exposed to multiple threats -most of them related to human activities- as they cross and use multiple habitats year-round, not because they are subjected to many different environmental conditions (to which they are -or should be- adapted). Please rephrase.

L53: I do not understand why you use “however”. I understand that phrase is taking about something completely different from the previous ones.

L54: Reference 3 does not support what is said.

L60: Reference 7 is about passerines. You should provide more references regarding the importance of stop-overs in species of marine or coastal habits.

L62-66. I do not agree with what is stated here. Determining the spatial distribution of seabirds during the migration period is a difficult task because birds they are generally located beyond the reach of the human eye. It is not obvious to me what you mean when saying “surveys of birds are usually limited to specific subsets of the total of the total migratory route”.L64: your statement is false for seabirds. Ring recoveries are not useful for identifying general movement patterns in seabirds, not only stop-over areas.

L68: accelerometers are not intended to track animal movements but behaviour, so they do not contribute to know at-sea distribution of seabirds.

L69: I do not agree. Light-level geolocators are the most extensively used devices to track migratory movements of seabirds, and the cost is not a general problem with them. Indeed, geolocators have the advantages of lilted weight, low price and animal-welfare, easy attachment, so high sample sizes are affordable. The real problem is what you explain in the next phrase. Even good sample sizes could be unrepresentative of seabird colony size (Soanes et al. (2013) How many seabirds do we need to track to define home‐range area? Journal of Applied Ecology, 50(3), 671-679; Thaxter, et al. (2017) Sample size required to characterize area use of tracked seabirds. The Journal of Wildlife Management, 81(6), 1098-1109).

L75: references are in wrong format. Moreover, both references do not refer to seabirds, even though you are already focused on seabirds since the first paragraph. Please use more appropriate references (i.e. regarding seabirds).

L76: format of references is wrong.

L77-78: Electronic devices are intended to provide individual tracking and thus get information at individual level. Taking this in mind, they can allow indeed long-term monitoring, and individuals of different species have already been tracked for more than a decade. In the case of seabirds and geolocators, devices can also be replaced annually, so as long as individuals are alive and can be recaptured, long term tracking can be carried out. Last, a major advantage of individual tracking is the fact that you know individual origin (i.e. colony, population). So, the weakness you state for tracking devices is inconsistent. Moreover, the weakness you argue about electronic devices also applied to census surveys, as birds counted at coastal points or vessel surveys are of unknown origin, and even species identification could be difficult for some species (as Balearic and Mediterranean shearwater). Instead of looking for weakness of tracking methods, you should highlight the strength of census methods (e.g. take an overview picture despite unknowing ages or colonies of origin). You should also comment a major weakness of census method for seabirds, which is the impossibility to count birds when they use areas out of the human sight during wintering or migration.

L84-91: I think here you did well in relating the strength of citizen science projects- I would suggest you to combine these lines with the previous paragraph, once removed/resolved the issues I said before. May this reference is also of interest for citation: Coxen, C. L., Frey, J. K., Carleton, S. A., & Collins, D. P. (2017) Species distribution models for a migratory bird based on citizen science and satellite tracking data. Global ecology and conservation, 11, 298-311. Last, you should introduce Trektellen here and comment if some other paper has use this datasource before, as I see this is a really good point of your paper.

L92-96: These lines may be in a different paragraph and be complemented with information about techniques for species distribution modelling. As I already said, many algorithms are available to model and predict species distribution and abundance, and some methods even integrated multiple algorithm. Thus, there is room here to give readers a bit introduction about this issue. I would include information on satellite-derived features as second part in the paragraph, as such features are the predictors used in the models.

L100: there is a typo after “shearwaters”.

L108-109: Please provide a reference. Which unused areas?

L114: replace “useful data” by “useful source of data”

L116: replace “these datasets” by “citizen science data”

L118: “most likely pelagic habitats”. This is inconsistent. You said in L112 the species occurs mainly in shallow coastal waters. Also, observation from coast only frequently allow to count birds in coastal waters. So identify pelagic habitats used during migration seems unfeasible.

L120: I believe the aim was to evaluate the performance, not to determine it

L121: replace “seabirds” by “Balearic shearwaters”.

L122: remove “of these and other species”. Also, you could include at the end something like “and discuss the potential of citizen science data for seabird conservation”.

L134-136: I suggest you include this in the next section.

L147: Ref. 32 is related to a different species, remove it.

L148: you should cite this paper, as one of the first describing environmental features to predict the presence of Balearic shearwaters: Louzao, M.et al. (2006) Oceanographic habitat of an endangered Mediterranean procellariiform: implications for marine protected areas. Ecological applications, 16(5), 1683-1695.

L33: this reference is from a symposium in 1993. Please, try to provide more appropriate references. Lot of really good work has been done in the last 20 years with Balearic shearwaters, even addressing the specific question you mention in L10-151. Just two examples:

• Jones, A. R., Wynn, R. B., Yésou, P., Thébault, L., Collins, P., Suberg, L., ... & Brereton, T. M. (2014). Using integrated land-and boat-based surveys to inform conservation of the Critically Endangered Balearic shearwater. Endangered Species Research, 25(1), 1-18.

• Wynn, R. B., Josey, S. A., Martin, A. P., Johns, D. G., & Yésou, P. (2007). Climate-driven range expansion of a critically endangered top predator in northeast Atlantic waters. Biology Letters, 3(5), 529-532.

L151: replace “pelagic and demersal” by “pelagic but also demersal”.

L151: You start the phrase referring to Balearic shearwater’s diet. I guess where you say “but it also feeds” are referring to the species, not the diet. Let’s say: “Balearic shearwater’s diet includes small pelagic but also demersal fish, frequently obtained from trawling discards. The species can eventually feed on plankton and macrozooplankton, specifically krill”.

Material & Methods.

L166: I recommend changing “observatories” by sighting points for clarity.

L171-172: May 1st – August 30th.

L179: Ref. 39 appears two times

L187: I find necessary to provide also a reference on shearwaters, at least a closely related species. Dias, M. P., Granadeiro, J. P., & Catry, P. (2012). Do seabirds differ from other migrants in their travel arrangements? On route strategies of Cory’s shearwater during its trans-equatorial journey. PLoS One, 7(11), e49376.

L187: you say “fluctuations in migration are closely related to changes in food resource”. What are you referring to? Do you mean bird abundance during migration? Do you mean fluctuations in areas used (location) during migration? Do you mean fluctuations in dates? Please clarify.

L192-193. I find this phrase unnecessary, and the reference provided not appropriate. Papers on seabird research usually indicate. I would suggest rephrasing to something like “We used chlorophyll concentration (Chla, measured in mg.m-3) as a proxy of marine productivity [ref]. We download satellite-based monthly products at 4 km spatial resolution for 2003-2017 period”. Ref. is Wakefield, E. D., Phillips, R. A., & Matthiopoulos, J. (2009). Quantifying habitat use and preferences of pelagic seabirds using individual movement data: a review. Marine Ecology Progress Series, 391, 165-182.

L195-198: This is an important methodological issue. I am going to use this moment to comment my concern about the unbalance in temporal representativity of predictors. You only use/have two years of fishing vessels distribution…I wonder how the unevenly representativity of predictor can affect Random Forest accuracy and predictions….

L202: I find necessary to cite at the end of the phrase “…modulating migratory behaviour” the reference: González-Solís, J., Felicísimo, A., Fox, J. W., Afanasyev, V., Kolbeinsson, Y., & Muñoz, J. (2009). Influence of sea surface winds on shearwater migration detours. Marine Ecology Progress Series, 391, 221-230.

L203: First reference in incorrect format, and also I find this reference no proper here. But again you have many to pick up one:

• Catry, P., Dias, M. P., Phillips, R. A., & Granadeiro, J. P. (2011). Different means to the same end: long-distance migrant seabirds from two colonies differ in behaviour, despite common wintering grounds. PLoS One, 6(10), e26079.

• Dias, M. P., Granadeiro, J. P., & Catry, P. (2013). Individual variability in the migratory path and stopovers of a long-distance pelagic migrant. Animal Behaviour, 86(2), 359-364.

• Dias, M. P., Granadeiro, J. P., & Catry, P. (2012). Do seabirds differ from other migrants in their travel arrangements? On route strategies of Cory’s shearwater during its trans-equatorial journey. PLoS One, 7(11), e49376.

But you can also cite this one that found out quite different results with regard to what you were saying:

• Dell’Ariccia, G., Benhamou, S., Dias, M. P., Granadeiro, J. P., Sudre, J., Catry, P., & Bonadonna, F. (2018). Flexible migratory choices of Cory’s shearwaters are not driven by shifts in prevailing air currents. Scientific reports, 8(1), 3376.

L203: Update [29] in Bibliography as this paper is already published.

I find that text in pages 10-11 could be reduced and summarize drastically to ease reading….Yo may included details in Supplementary Material. I think you give too much details about predictors used but could be omitted simply citing references were each predictor has been used before.

Regarding Table 1:

Indicate in the “Total” row that it is the response/target variable.

Batim: remove year

Since the work is based on citizen science, I would expect to find “day of the week” in the table of predictors, as volunteers usually go birding on weekend. Another important detail to me is that first paragraph of Results section should be moved to Material and Methods, as an exploratory analysis is part of methods and drives your next methodological steps. Idem for second paragraph in Results.

Another point to highlight in this point is that you do not show any data regarding sampling effort of the citizen science project. The consistency in these projects is key for data quality and even though RF can deal with this issue, it would be informative to see a plot with sampling effort (for example days of observation) at the 123 observatories over time.

Statistical analysis

I recommend starting this section clearly indicating the structure of your data matrix, i.e. clearly defining whether each row is an observatory point, a session (i.e. specific day) from an observatory point, or whatever.

L60: I would recommend citing references already published. If it is not possible, at least provide some extended explanation to understand your choice without the need to find a not yet published paper.

L243: It is not clear to me why you log-transformed the abundance data. Please clarify it.

L246-256: You should provide more details about tuning parameters of RF. Please clearly state the total sample size in the data matrix used as input in the models (number of rows, number of columns), the number of trees you grew up, the number of predictors you used for each tree, and the number of samples you select in each tree. It is not clear if you are using cross-validation, please try to clarify your explanations in L253.

L249: You refer RMSE without defining its meaning before. Please do explain briefly to the readers why you used Root Men Square Error for parameter tuning.

I would expected to see the confusion matrix (predicted vs observed) to generally evaluate accuracy of RF models. Could you include it?

L257: A bit confusing...May be rephrase: “Relative importance of the variables used for predicting shearwater abundance…”

Regarding Variable Importance, considering most readers of this paper will be ecologists from outside computing science, I would recommend to use a metric (and plots) easier to follow and understand. Ranking variables with the Gini Index or Mean Decrease Accuracy and showing a barplot illustrating the ranking would be much more understandable than plots provided by randomForestExplainer. In my opinion, when predictors overlap in the ranking the visualization provided by this R package is not so effective, as shown in figures 3a and 3b.

Results

As I said before, I think the first and the second paragraph of Results section should be moved to Material and Methods, as an exploratory analysis is part of methods and drives your next methodological steps.

L276-277: The way this sentence is written is somehow confusing, as you only run two models (pre & post-breeding migration), but from the text it seems as you run a lot. As building a RF model implies using different training data subsample, I recommend removing this from the sentences “among the models built with different training data subsamples”.

L308-310. This result could be due to a temporal bias in sampling effort from observatories. You should show, at least, the number of days of observation per Latitude and Year to support this result.

Fig. 5, figure caption: I think it is more properly as: “Predictions of Balearic shearwater spatial distribution and abundance during migration”

Discussion

L349-352: I understand Big Data is trending topic…But this paper is really far from using Big Data, so I recommend to avoid the use of this term.

L358: My suggestion: “allow to identify general spatio-temporal patterns as data come from different individuals belonging to several populations”.

You should discuss in this part the weakness of data coming from volunteering programmes, such as temporal gaps, uneven sampling effort in space and time, etc. particularly in L353 where you indicate “its characteristic noise” but do not explain anything more.

L361-365: This phrase is too long, 5 lines…. Please rephrase or split it.

L368: RF are not good to forecast out of the range of input data, so future projections based on this could be unreliable… depending on input data. Moreover, I think this issue (I mean future projections) is out of the scope of this paper.

L371: ref [74] is about ducks. I would say there are more appropriates references to cite here related with seabirds.

L379: format of references incorrect.

L382: Be consistent with therm used. Replace volunteer data by scitizen science data.

L393: what do you mean to “larger variability observed”? which result support this?

L395-397: This phrase is not clear. Please rephrase it.

L422: this is related to some clarifications needed that I mentioned before.

L433-448. I found Conclusions section clear and nice

Bibliography:

There are typos spread over the section.

6. PLOS authors have the option to publish the peer review history of their article (what does this mean?). If published, this will include your full peer review and any attached files.

Reviewer #1: No

---

## [Author Response · Author response to Decision Letter 0]

21 Feb 2020

22nd February 2020

PONE-D-19-29580

Citizen science and machine learning for predicting spatio-temporal patterns in seabird migration

David Hyrenbach, Ph.D.

Academic Editor

PLOS ONE

PLOS ONE

Dear Dr. Hyrenbach,

We sincerely thank you for allowing us to resubmit our paper. Following your instructions, we resubmit the manuscript titled “Citizen science and machine learning for predicting spatio-temporal patterns in seabird migration” (PONE-D-19-29580), now retitled “Citizen science for predicting spatio-temporal patterns in seabird abundance during migration”, after fully addressing all the concerns raised by you and the reviewer. 

Changes done throughout the text have been recorded, via track-changes, in a document enclosed with this re-submission. Please find below the detailed explanation of the modifications we have made in response to the concerns raised. A copy of the document with the track changes feature turned off is also attached. Lines indicated below and within the document with the track changes feature turned on, referred to the document with the track changes feature turned off. 

We hope that you will find this revised version suitable for publication in PLOS ONE.

Yours faithfully,

Beatriz Martín on behalf of all the authors

PONE-D-19-29580

Citizen science and machine learning for predicting spatio-temporal patterns in seabird migration

PLOS ONE

Dear Dr Martin,

Thank you for submitting your manuscript to PLOS ONE. After careful consideration, we feel that it has merit but does not fully meet PLOS ONE’s publication criteria as it currently stands. Therefore, we invite you to submit a revised version of the manuscript that addresses the points raised during the review process.

i apologize for the long handling time, but securing reviewers proved very difficult. Because I was only able to secure one review, I did another review of the ms, and I concluded that it needed substantial revisions before it could be published. I forward an annotated version with inserted comments, edits, and questions. In particular, I urge the authors to:

1) revise the introduction to remove redundancies / and focus focus on the main goals.

Authors: Done. See changes made throughout the Introduction section.

2) revise the introduction and discussion to ensure the limitations of citizen science data are acknowledged\\. currently, the ms highlights the limitations of tracking and vessel-based surveys, but there are many similar limitations to citizen science data: uncertainty of identification, uncertainty of counts, uneven coverage and lack of systematic effort, and lack of knowledge of the colony provenance of the birds sighted at sea.

Authors: Done. See changes made in the Introduction and Discussion sections.

3) revise the methods section (and augment with supplementary tables) to provide readers with a sense of how the sighting / fishery / environmental datasets were cleaned / processed. in particular, address the uncertainties associated with the species identifications, and explicitly explain how the fishery data were subsetted by sector / fishery / target.

Authors: New figures have been added both in Methods (see new Figure 1) and in a new Supplementary Material section (Figures S1, S2 and S3), showing the temporal and spatial representation of the data, as well as the count effort distribution over the days of the week (as requested by the reviewer#1, see answer to comments below regarding this issue). In addition, the results of the collinearity assessment are also now reported in the Supplementary Material. Balearic shearwater generates much attention both from amateur and expert ornithologists thus it is usually well-known species and correctly identified when sighted. However, we cannot totally exclude species identification errors among the different sources of variation associated with citizen science data. The previous is now stated in the text (see lines 167-170). Regarding the data on fisheries, JRC provides a detailed map of high intensity fisheries areas in 2014-2015 in Europe from tracking data of fishing vessels. The map allows to identify which are the areas where they fish more frequently. Vessel tracking data was derived from the open source Automatic Identification System (AIS), allowing to analyse the relation between fishing communities and fishing areas at high spatial resolution across Europe. Data used to build the map consist of 150 million positions from European fishing vessels above 15 m in length. This map was created to be used by scientist and experts for fisheries management and fisheries research both from an environmental and a socio-economic perspective. We have now clarified all that in the text.

4) furthermore, please provide evidence that the authors had cleared the terms of use of these publicly available sighting data, and are thus able to publish these datasets. moreover, explicitly state how many records / birds were included in each dataset, for the pre and post breeding migration. finally, explicitly state whether "zero" counts were included, if standardized surveys (or sightings of other species were reported, but no Balearic Shearwatyer sightings were reported).

Authors: Terms of use of eBird database have been cleared (https://www.ala.org.au/wp-content/uploads/2015/02/Terms_of_Use.v3.pdf). Trektellen is a public database of migration / seawatch counts and ringing results (see additions into the Acknowledge section). On the other hand, only records attributed to Balearic shearwaters were analysed. Both systematic and opportunistic data were analysed. Although it was possible to differentiate opportunistic and systematic surveys within the dataset, we opted for keeping all records for our analysis in order to test the robustness of the modelling approach in case our methods will be extended to other species for which this information is not available. This has now clarified in the text (see lines 171-174). In addition, we modelled species abundance with only abundance-given-presence (Pearce & Boyce 2006), thus zeros were not included in the data matrix (see lines 192-193). On the other hand, after the removal of data recorded before 2005 and after 2017, 7,492 and 4,690 observations (i.e., rows in the data matrix) remained for the pre-breeding and post-breeding migration periods, respectively. Number of records (as well as number of birds) considered in the analysis after removal are reported in the Results section (see lines 266-272).

5) the methods also need to be strengthened, since the ms refers to several exploratory analyses (daily abundances used to censor the study period and co-linearity of explanatory variables), but no results are provided. I urge the authors to provide supplementary tables with these results, so the readers can evaluate what was done.

Authors: Please see changes made throughout the Methods section as well as the new Supplementary Material attached to the manuscript including new figures reporting these issues.

6) the results also need to be strengthened, to ensure tests are clearly reported and evaluated. in particular, please use the comments added to the ms to address the comparison of the model R-squared values, and the correlations or modeled / observed abundances. finally, please justify the small number of iterations used (25) and the use of means +-/ 95% CI for some of the estimated parameters, rather than medians (25% - 75%).

Authors: We have followed the recommendations made throughout the document and we have clarified the comparisons and the tests performed (please see answers to comments embedded in the manuscript with track changes on). Regarding iterations and 95% CI, as it is stated in the Methods section, any random forest model apply bagging (i.e., bootstrap aggregating) to sub-sample the data that are used for training, thus each new tree is fit from a bootstrap sample of the training observations zi=(xi,yi). Therefore, using 25 repetitions of this bootstrap resampling actually means a lot of different trees to be built. Large data samples tend to approximate the normal distribution thus we think that the use of 95% CI is correct.

7) The discussion and conclusions were clear and concise, but they are difficult to evaluate , given the paucity of details in the methods / results. Finally, I urge the authors to carefully consider how they identify "key" areas, and to consider that they are only using counts; there are no behavioral data (eg., feeding versus transiting). Namely, would a high flux area (e.g., Gibraltar Strait) with lower counts be more sensitive to impacts than a foraging area with higher counts but lower flux? 

Authors: Details have added in Methods and Results section following the recommendations of both the editor and the reviewer. Regarding the key areas, although we are analysing no behavioural data, to some extent, we can infer feeding and transiting areas from variable interactions. For instance, locations with tailwinds, high abundance level and low chlorophyll concentration can be interpreted as locations with high flux of birds, in contrast to locations with moderate abundance, no winds or moderate winds and high chlorophyll concentration levels. We have added the previous information in the new version of the manuscript (see lines 388-392). In addition, for the sake of clarity, we have moved the content related to this issue contained at the end of the Results section into the Discussion section, because we actually were discussing our results in relation to previous findings (see lines 392-399).

We would appreciate receiving your revised manuscript by Feb 22 2020 11:59PM. To enhance the reproducibility of your results, we recommend that if applicable you deposit your laboratory protocols in protocols.io, where a protocol can be assigned its own identifier (DOI) such that it can be cited independently in the future. For instructions see: http://journals.plos.org/plosone/s/submission-guidelines#loc-laboratory-protocols

• A rebuttal letter that responds to each point raised by the academic editor and reviewer(s). This letter should be uploaded as separate file and labeled 'Response to Reviewers'.

• A marked-up copy of your manuscript that highlights changes made to the original version. This file should be uploaded as separate file and labeled 'Revised Manuscript with Track Changes'.

• An unmarked version of your revised paper without tracked changes. This file should be uploaded as separate file and labeled 'Manuscript'.

We look forward to receiving your revised manuscript.

Kind regards,

David Hyrenbach, Ph.D.

Academic Editor

PLOS ONE

Journal Requirements:

'This study is part of a research project (“Environmental factors determining the interannual variation in the migration of Balearic and Scopoli’s shearwaters in the Mediterranean”, 2018-2019) which has been partly financed by the Annual Programme of Grants of the Instituto de Estudios Ceutíes (IEC, Autonomous City of Ceuta, Spain), years 2018-2019.'

 Authors: Done.

Reviewers' comments:

Reviewer's Responses to Questions

Comments to the Author

1. Is the manuscript technically sound, and do the data support the conclusions?

Reviewer #1: Yes

2. Has the statistical analysis been performed appropriately and rigorously? 

Reviewer #1: Yes

3. Have the authors made all data underlying the findings in their manuscript fully available?

Reviewer #1: Yes

4. Is the manuscript presented in an intelligible fashion and written in standard English?

Reviewer #1: Yes

5. Review Comments to the Author

Reviewer #1: In this manuscript, authors present a novel work to evaluate whether data from a long-term monitoring programme carried out by volunteers, i.e., a citizen science project, can be used to assess bird migratory patterns. They focused on an endangered seabird species endemic to the Mediterranean Sea that migrates to the Atlantic out of the breeding period. Authors relied on supervised machine learning technics to predict bird abundance, building up the models from data gathered at a network of bird census stations. Specifically, they used random forest regression models to evaluate the usefulness of citizen science data in predicting spatial and temporal abundance based on a set of predictors, including many satellite-derived features, reaching an accuracy of about 70%. Authors concluded that combining long-term citizen science with predictive modelling can be a reliable tool to assist long-term monitoring of sensitive species, to identify important areas and/or to detect trends.

The paper is well structured. The study is accurate and provides an interesting approach. As general comments, the major strength of this study relies on the source data used. Trektellen is an online database recording opportunistic seabird counts carried out by volunteers across Europe. Despite its potential, very few papers have made use of this long-term database. As a counterpart, there are also some weaknesses in the work that should be amended for publication. My main concern in this regard is that authors tend to overstate along the manuscript the importance of random forest (RF) to predict spatial and temporal abundance. RF is a really powerful algorithm (I personally really love it) based on an ensemble of decision trees, but its use is not new in the context of species distribution/abundance modelling (e.g. Oppel et al. (2012) Comparison of five modelling techniques to predict the spatial distribution and abundance of seabirds. Biological Conservation, 156, 94-104). In fact, recent approaches for species distribution modelling tend to build up a final output based on an ensemble of several algorithms (a.k.a. Ensemble Ecological Niche Models), including RF among them. See the “ssdm” package in R for details (Schmitt et al. (2017) ssdm: An r package to predict distribution of species richness and composition based on stacked species distribution models. Methods in Ecology and Evolution, 8(12), 1795-1803; and also, e.g., Pereira et al. (2018) Using a multi-model ensemble forecasting approach to identify key marine protected areas for seabirds in the Portuguese coast. Ocean & Coastal Management, 153, 98-107.). 

Authors: We thanks the reviewer for the mentioned publications in relation to the subject of our study. In fact, Oppel et al. (2012) was cited and discussed in our manuscript (see lines 108, 345-351). However, this previous research found Balearic shearwater abundance to be extremely difficult to predict and it failed to provide reliable predictions on the spatial distribution of shearwater numbers. Moreover, their predictions were based on abundance and absence data. In contrast to this earlier modelling attempt based on ship transect data, our models, based on abundance given presence, provided strongly correlated predictions with the observed abundances and explained a significant proportion of the variance existing in the numbers of migrating shearwaters. 

In relation to the other studies mentioned by the reviewer (Schmitt et al. 2017; Pereira et al. 2018), we are afraid that we cannot applied the above-mentioned approaches (based on ecological niche modelling and Individual Species Distribution Models -SDM-) because in our study, we are not modelling occurrence but abundance (i.e., count data of Balearic shearwaters). In addition, presence-only SDMs models are based on correlations between species presence and environmental conditions, predicting the environmental suitability for a species, and not their realized distribution, leading to weak relationships between predicted presence and relative abundance (Gomes et al. 2018). By modelling abundance instead of occurrence, we can obtain predictions on real abundance that can provide more relevant information to delineate marine protected areas. In relation to models predicting animal occurrence, few studies have tried to model estimates of relative abundance, even when data are acquired through systematic surveys. In addition, modelling only abundance-given-presence approaches as that in our study have been a rare approach in ecological modelling (Pearce and Boyce 2006). All this further discussion has been included in the Discussion section of the manuscript (lines 343-351). In addition, we have clarified our approach both in the Introduction and Methods section. We have also changed the title of the manuscript, replacing the reference to the “Random Forest” analysis and highlighting the fact that we model abundance data. 

Reviewer #1: Apart from that, a major weakness of RF, just as in decision trees, is that they cannot extrapolate outside the range of the training data, and therefore great care must be taken when these algorithms are used to predict trends in the long-term, as those expected in the context of climate change. For these reasons, I think authors should attenuate the importance of using this machine learning algorithm in the paper, beginning with removing it from the title.

Authors: We agree with the reviewer#1 that Random forest, like other machine learning algorithms, are quite unreliable when extrapolating outside the range of the predictors’ values provided for training. We have removed mentions to future projections (see line 398 and lines 446) and added the specific validity of these models for understanding environmental changes within the range of the training data (see line 476). However, even with this limitation, as it is stated in the text, “trends in abundance related to changes in fisheries and/or in climate affecting the Balearic shearwater distribution can be inspected from a range of latitude and year values over the study period” (see lines 426-429).

Reviewer#1: Below I will refer to specific section and lines to provide comment:

As I have already commented, I would recommend changing the title, omitting reference to machine learning on it.

Authors: The title has been changed according to the answers given to the reviewer#1 comments above. Following the reviewer suggestion, we have removed “random forest” mention but we have highlighted the use of abundance data.

Reviewer: Abstract:

There is a lack of introduction. Please include at least one or two sentences about the general framework before stating what you did address.

Authors: Done. Please see lines 29-30.

Reviewer: Introduction:

In general, I think the introduction needs a bit of work. Some sentences are not explained well enough. The first paragraph is not very cohesive.

Authors: We have now re-written the Introduction section.

Reviewer: L49: Should be rephrased. Although I understand what authors mean, these concepts could be explained in a clearer way.

Authors: Done.

Reviewer: L52-53: I would say seabirds are threatened because they are exposed to multiple threats -most of them related to human activities- as they cross and use multiple habitats year-round, not because they are subjected to many different environmental conditions (to which they are -or should be- adapted). Please rephrase.

Authors: Done.

Reviewer: L53: I do not understand why you use “however”. I understand that phrase is taking about something completely different from the previous ones.

Authors: This sentence has been now totally re-phrased.

Reviewer: L54: Reference 3 does not support what is said.

Authors: This reference was removed and replaced by Felis et al. 2019.

Reviewer: L60: Reference 7 is about passerines. You should provide more references regarding the importance of stop-overs in species of marine or coastal habits.

Authors: Done. We have removed this citation and we have replaced it by a new one on seabirds. Please see new citation (Felis et al. 2019). In addition, we have also replaced other citations by specific seabird references (see line 61, Guilford et al. 2009).

Reviewer: L62-66. I do not agree with what is stated here. Determining the spatial distribution of seabirds during the migration period is a difficult task because birds they are generally located beyond the reach of the human eye. It is not obvious to me what you mean when saying “surveys of birds are usually limited to specific subsets of the total of the total migratory route”.

Authors: We agree with the reviewer and we have re-phrased the sentence accordingly.

Reviewer: L64: your statement is false for seabirds. Ring recoveries are not useful for identifying general movement patterns in seabirds, not only stop-over areas.

Authors: We disagree with the reviewer. Traditional studies based on ringing recoveries or ocean sightings have proved valuable for identifying very general movement patterns, but fail to discriminate important localities at sea (please see the citation originally provided -Walker et al. 2013- and also Guilford et al. 2009, which is now also cited in the text).

Reviewer: L68: accelerometers are not intended to track animal movements but behaviour, so they do not contribute to know at-sea distribution of seabirds.

Authors: We agree with the reviewer thus we have removed this mention in the new version of text.

Reviewer: L69: I do not agree. Light-level geolocators are the most extensively used devices to track migratory movements of seabirds, and the cost is not a general problem with them. Indeed, geolocators have the advantages of lilted weight, low price and animal-welfare, easy attachment, so high sample sizes are affordable. The real problem is what you explain in the next phrase. Even good sample sizes could be unrepresentative of seabird colony size (Soanes et al. (2013) How many seabirds do we need to track to define home‐range area? Journal of Applied Ecology, 50(3), 671-679; Thaxter, et al. (2017) Sample size required to characterize area use of tracked seabirds. The Journal of Wildlife Management, 81(6), 1098-1109).

Authors: We agree with the reviewer thus we have re-phrased the sentence accordingly. In addition, both citations indicated by the reviewer have been included in the new version of the text.

Reviewer: L75: references are in wrong format. Moreover, both references do not refer to seabirds, even though you are already focused on seabirds since the first paragraph. Please use more appropriate references (i.e. regarding seabirds).

Authors: Done.

Reviewer: L76: format of references is wrong.

Authors: The format has been now corrected.

Reviewer: L77-78: Electronic devices are intended to provide individual tracking and thus get information at individual level. Taking this in mind, they can allow indeed long-term monitoring, and individuals of different species have already been tracked for more than a decade. In the case of seabirds and geolocators, devices can also be replaced annually, so as long as individuals are alive and can be recaptured, long term tracking can be carried out. Last, a major advantage of individual tracking is the fact that you know individual origin (i.e. colony, population). So, the weakness you state for tracking devices is inconsistent. Moreover, the weakness you argue about electronic devices also applied to census surveys, as birds counted at coastal points or vessel surveys are of unknown origin, and even species identification could be difficult for some species (as Balearic and Mediterranean shearwater). Instead of looking for weakness of tracking methods, you should highlight the strength of census methods (e.g. take an overview picture despite unknowing ages or colonies of origin). You should also comment a major weakness of census method for seabirds, which is the impossibility to count birds when they use areas out of the human sight during wintering or migration.

Authors: Done, please see lines 77-79.

Reviewer: L84-91: I think here you did well in relating the strength of citizen science projects- I would suggest you to combine these lines with the previous paragraph, once removed/resolved the issues I said before. May this reference is also of interest for citation: Coxen, C. L., Frey, J. K., Carleton, S. A., & Collins, D. P. (2017) Species distribution models for a migratory bird based on citizen science and satellite tracking data. Global ecology and conservation, 11, 298-311. Last, you should introduce Trektellen here and comment if some other paper has use this datasource before, as I see this is a really good point of your paper.

Authors: We have followed all the suggestions made by the reviewer in this specific comment.

Reviewer: L92-96: These lines may be in a different paragraph and be complemented with information about techniques for species distribution modelling. As I already said, many algorithms are available to model and predict species distribution and abundance, and some methods even integrated multiple algorithm. Thus, there is room here to give readers a bit introduction about this issue. I would include information on satellite-derived features as second part in the paragraph, as such features are the predictors used in the models.

Authors: We have included additional information on modelling techniques and on the relevance of our modelling approach both in the Introduction and in the Discussion sections. However, for the sake of clarity, we have kept this information after the information on satellite-derived features. See lines 125-131 and also answers to comments above.

Reviewer: L100: there is a typo after “shearwaters”.

Authors: The comma has now been removed.

Reviewer: L108-109: Please provide a reference. Which unused areas?

Done. We referred to unused areas non-detected in the previous studies. See Meier et al. 2015 which is now cited in the text.

Reviewer: L114: replace “useful data” by “useful source of data”

Following the other reviewer suggestion, “other useful data” have been replaced by “other records”.

Reviewer: L116: replace “these datasets” by “citizen science data”

Authors: This sentence has been rephrased following the other reviewer suggestions, thus “these datasets” has now been removed.

Reviewer: L118: “most likely pelagic habitats”. This is inconsistent. You said in L112 the species occurs mainly in shallow coastal waters. Also, observation from coast only frequently allow to count birds in coastal waters. So identify pelagic habitats used during migration seems unfeasible.

Authors: “pelagic” has been replaced by “marine”.

Reviewer: L120: I believe the aim was to evaluate the performance, not to determine it

Authors: We agree with the reviewer observation, thus “determine” has been replaced by “evaluate”.

Reviewer: L121: replace “seabirds” by “Balearic shearwaters”.

Authors: Done.

Reviewer: L122: remove “of these and other species”. Also, you could include at the end something like “and discuss the potential of citizen science data for seabird conservation”.

Authors: Done.

Reviewer: L134-136: I suggest you include this in the next section.

Authors: Done.

Reviewer: L147: Ref. 32 is related to a different species, remove it.

Authors: Done.

Reviewer: L148: you should cite this paper, as one of the first describing environmental features to predict the presence of Balearic shearwaters: Louzao, M.et al. (2006) Oceanographic habitat of an endangered Mediterranean procellariiform: implications for marine protected areas. Ecological applications, 16(5), 1683-1695.

Authors: Done.

Reviewer: L33: this reference is from a symposium in 1993. Please, try to provide more appropriate references. Lot of really good work has been done in the last 20 years with Balearic shearwaters, even addressing the specific question you mention in L10-151. Just two examples:

• Jones, A. R., Wynn, R. B., Yésou, P., Thébault, L., Collins, P., Suberg, L., ... & Brereton, T. M. (2014). Using integrated land-and boat-based surveys to inform conservation of the Critically Endangered Balearic shearwater. Endangered Species Research, 25(1), 1-18.

• Wynn, R. B., Josey, S. A., Martin, A. P., Johns, D. G., & Yésou, P. (2007). Climate-driven range expansion of a critically endangered top predator in northeast Atlantic waters. Biology Letters, 3(5), 529-532.

Authors: Done.

Reviewer: L151: replace “pelagic and demersal” by “pelagic but also demersal”.

Authors: Done.

Reviewer: L151: You start the phrase referring to Balearic shearwater’s diet. I guess where you say “but it also feeds” are referring to the species, not the diet. Let’s say: “Balearic shearwater’s diet includes small pelagic but also demersal fish, frequently obtained from trawling discards. The species can eventually feed on plankton and macrozooplankton, specifically krill”.

Authors: Done.

Material & Methods.

Reviewer:L166: I recommend changing “observatories” by sighting points for clarity.

Authors: Done.

Reviewer:L171-172: May 1st – August 30th.

Authors: Done.

Reviewer:L179: Ref. 39 appears two times

Authors: We have removed the duplication.

Reviewer:L187: I find necessary to provide also a reference on shearwaters, at least a closely related species. Dias, M. P., Granadeiro, J. P., & Catry, P. (2012). Do seabirds differ from other migrants in their travel arrangements? On route strategies of Cory’s shearwater during its trans-equatorial journey. PLoS One, 7(11), e49376.

Authors: Done.

Reviewer:L187: you say “fluctuations in migration are closely related to changes in food resource”. What are you referring to? Do you mean bird abundance during migration? Do you mean fluctuations in areas used (location) during migration? Do you mean fluctuations in dates? Please clarify.

Authors: We referred to bird abundance during migration. This has now been clarified in the new version of the text.

Reviewer:L192-193. I find this phrase unnecessary, and the reference provided not appropriate. Papers on seabird research usually indicate. I would suggest rephrasing to something like “We used chlorophyll concentration (Chla, measured in mg.m-3) as a proxy of marine productivity [ref]. We download satellite-based monthly products at 4 km spatial resolution for 2003-2017 period”. Ref. is Wakefield, E. D., Phillips, R. A., & Matthiopoulos, J. (2009). Quantifying habitat use and preferences of pelagic seabirds using individual movement data: a review. Marine Ecology Progress Series, 391, 165-182.

Authors: Done.

Reviewer:L195-198: This is an important methodological issue. I am going to use this moment to comment my concern about the unbalance in temporal representativity of predictors. You only use/have two years of fishing vessels distribution…I wonder how the unevenly representativity of predictor can affect Random Forest accuracy and predictions….

Authors: Discard availability may influence the at-sea distribution of shearwaters. As a proxy of food availability (both discards and fishes) we used information on fisheries. This information was inferred from the distribution of fishing vessels between 2014 and 2015, sourced by the JRC Data Catalogue. This product identifies which are the areas where fishing is more frequent. Vessel tracking data was derived from the Automatic Identification System (AIS), an open source data allowing to analyse the relation between fishing communities and fishing areas at high spatial resolution across Europe. Specifically, data used to build the map consist of 150 million positions from European fishing vessels above 15 m in length. In spite of its limited temporal coverage, the main strength of this dataset is its fine spatial resolution. This proxy on food availability in spatial terms, however, is complemented in our analysis with the high temporal resolution in marine productivity provided by chlorophyll concentration.

Reviewer:L202: I find necessary to cite at the end of the phrase “…modulating migratory behaviour” the reference: González-Solís, J., Felicísimo, A., Fox, J. W., Afanasyev, V., Kolbeinsson, Y., & Muñoz, J. (2009). Influence of sea surface winds on shearwater migration detours. Marine Ecology Progress Series, 391, 221-230.

Authors: Done.

Reviewer:L203: First reference in incorrect format, and also I find this reference no proper here. But again you have many to pick up one:

• Catry, P., Dias, M. P., Phillips, R. A., & Granadeiro, J. P. (2011). Different means to the same end: long-distance migrant seabirds from two colonies differ in behaviour, despite common wintering grounds. PLoS One, 6(10), e26079.

• Dias, M. P., Granadeiro, J. P., & Catry, P. (2013). Individual variability in the migratory path and stopovers of a long-distance pelagic migrant. Animal Behaviour, 86(2), 359-364.

• Dias, M. P., Granadeiro, J. P., & Catry, P. (2012). Do seabirds differ from other migrants in their travel arrangements? On route strategies of Cory’s shearwater during its trans-equatorial journey. PLoS One, 7(11), e49376.

But you can also cite this one that found out quite different results with regard to what you were saying:

• Dell’Ariccia, G., Benhamou, S., Dias, M. P., Granadeiro, J. P., Sudre, J., Catry, P., & Bonadonna, F. (2018). Flexible migratory choices of Cory’s shearwaters are not driven by shifts in prevailing air currents. Scientific reports, 8(1), 3376.

Authors: We have added two of the citations suggested by the reviewer supporting our statement as well as the one showing opposite results. 

Reviewer:L203: Update [29] in Bibliography as this paper is already published.

Authors: Done.

Reviewer: I find that text in pages 10-11 could be reduced and summarize drastically to ease reading….Yo may included details in Supplementary Material. I think you give too much details about predictors used but could be omitted simply citing references were each predictor has been used before.

Authors: Done. See the new Supplementary Material section (specifically, the Supplementary Methods section).

Reviewer: Regarding Table 1: 

Indicate in the “Total” row that it is the response/target variable.

Authors: Following the editor suggestion, response variable has been removed from Table 1 in the present version of the manuscript.

Reviewer: Batim: remove year

Authors: Done. See also the other non-environmental predictors removed according to suggestions made by the editor.

Reviewer: Since the work is based on citizen science, I would expect to find “day of the week” in the table of predictors, as volunteers usually go birding on weekend. 

Authors: Day of the week should not affect the magnitude of the response variable (abundance of shearwaters) thus we think it should not be included as an additional predictor of shearwater abundance. However, we agree with the reviewer that it could affect the temporal representation of the data. In this sense, we have added a new Figure (see Figure S2 in the Supplementary Material) showing that the datasets considered in our study contain a sufficient sample of observations in different days over the week.

Reviewer: Another important detail to me is that first paragraph of Results section should be moved to Material and Methods, as an exploratory analysis is part of methods and drives your next methodological steps. Idem for second paragraph in Results.

Authors: We disagree with the reviewer and we think that even if exploratory analysis, these are preliminary results that must be included in the Results section thus we have kept the structure of this section as it was.

Reviewer: Another point to highlight in this point is that you do not show any data regarding sampling effort of the citizen science project. The consistency in these projects is key for data quality and even though RF can deal with this issue, it would be informative to see a plot with sampling effort (for example days of observation) at the 123 observatories over time.

Authors: As our response variable are daily counts, number of records reported in the Results section is actually the total number of sampling days considered in the analysis. Total number of birds sighted during this sampling days is also reported in the Results section (see also answers to the editor comments above). In addition, regarding the sampling effort over time, we now provide the records conducted in different months over the study period considered in our research, as well as in different days of the week (see Figure S1 and S2 in the Supplementary Material; see also answers to comments above).

Statistical analysis

Reviewer: I recommend starting this section clearly indicating the structure of your data matrix, i.e. clearly defining whether each row is an observatory point, a session (i.e. specific day) from an observatory point, or whatever.

Authors: Done.

Reviewer: L60: I would recommend citing references already published. If it is not possible, at least provide some extended explanation to understand your choice without the need to find a not yet published paper.

Authors: Unfortunately, the manuscript cited is still under review thus we cannot provide the reference to a published paper. However, following the suggestion of the reviewer, we have provided an extended explanation of the results obtain in the mentioned research. Please, see lines 216-233.

Reviewer: L243: It is not clear to me why you log-transformed the abundance data. Please clarify it.

Authors: As it is stated in the text, abundance of shearwaters are count data that follow a Poisson distribution. Prior to build our models we log-transformed the abundance data in order to improve the accuracy of the estimated effects reducing the potential bias and increasing the predictive power of the model. We have included this clarification in the new version of the manuscript.

Reviewer: L246-256: You should provide more details about tuning parameters of RF. Please clearly state the total sample size in the data matrix used as input in the models (number of rows, number of columns), the number of trees you grew up, the number of predictors you used for each tree, and the number of samples you select in each tree. It is not clear if you are using cross-validation, please try to clarify your explanations in L253.

Authors: Sample size of the shearwater observations (i.e., rows in the data matrix) is provided in the Results section (see lines 278-282). As in any usual regression analysis, the number of columns in the data matrix matched the number of predictors (see Table 1). All the trees were built using the total set of predictors described in Table 1. We assessed the performance of the models based on the Out- of-bag (OOB) error. The random forest model was trained using bootstrap aggregation, where each new tree was fit from a bootstrap sample of the training observations zi=(xi,yi). The OOB error is the average error for each zi calculated using predictions from the trees that do not contain zi in their respective bootstrap sample (Hastie et al. 2009). This allows the model to be fit and validated whilst being trained, thus no additional cross-validation was required. Number of trees assessed ranged between 100 and 500. For tuning the parameters of the model, we applied a grid search method, thus we evaluated the model over different combinations of parameters included in the grid (values ranging between 1-15 at one unit intervals for mtry parameter).

For the sake of clarity, we have re-written this section and all this explanations and the above-mentioned citation have been now included in the Methods section. 

T. Hastie, R. Tibshirani and J. Friedman, “The Elements of Statistical Learning 2nd: Data Mining, Inference, and Prediction”, p592-593, Springer, 2009.

Reviewer: L249: You refer RMSE without defining its meaning before. Please do explain briefly to the readers why you used Root Men Square Error for parameter tuning.

I would expected to see the confusion matrix (predicted vs observed) to generally evaluate accuracy of RF models. Could you include it?

Authors: RMSE (Root Mean Square Error) is the average difference between the observed known values of the outcome and the predicted value by the model. It is not solely a measure for parameter tuning but, on the whole, a usual measure used for assessing the accuracy of random forest regression models. The lower the RMSE value, the better the model. Further clarifications regarding this issue have been now included in the Methods section. 

Regarding confussion matrix, we cannot provide it because we are modelling abundance (i.e., count data; see new lines added: 192-193), thus we are not calibrating random forest classification trees (models where the response variable is binary) but random forest regression trees (models where the response variable is continuous). Therefore, as it is stated in the text, we inferred the correspondence between observed and predicted values from RMSE. 

Reviewer: L257: A bit confusing...May be rephrase: “Relative importance of the variables used for predicting shearwater abundance…”

Authors: Done.

Reviewer: Regarding Variable Importance, considering most readers of this paper will be ecologists from outside computing science, I would recommend to use a metric (and plots) easier to follow and understand. Ranking variables with the Gini Index or Mean Decrease Accuracy and showing a barplot illustrating the ranking would be much more understandable than plots provided by randomForestExplainer. In my opinion, when predictors overlap in the ranking the visualization provided by this R package is not so effective, as shown in figures 3a and 3b.

Authors: As we explained above, we are modelling random forest regression trees, thus we cannot apply the measures suggested by the reviewer, since Gini Inces and Mean Decrease Accuracy are measures of node impurity in classification Random Forest. In addition, these ranks rely on the choice of a performance measure, although measures of prediction performance are not unique. To solve this, Ishwaran et al. (2010) developed a method called minimal depth (MD) that simply determines variable importance by the position of the variables in the decision trees and thus, is only based on the decision tree structures. The idea is that, variables that tend to split close to the root node should have more importance in prediction. We think that our approach, even more complex, provides a better understanding of the relative variable importance and, therefore, it should be kept, since it is not only a matter of plotting but a matter of the way we quantified the variable importance.

Results

Reviewer: As I said before, I think the first and the second paragraph of Results section should be moved to Material and Methods, as an exploratory analysis is part of methods and drives your next methodological steps.

Authors: See comment above. In our experience (as researchers writing papers and as reviewers of scientific publications) even if they are exploratory, all analysis conducted are expected in the Results section, since the Methods section is only a description of the procedures that were applied to obtained the results. Therefore, we decided to keep the structure of this section as it was.

Reviewer: L276-277: The way this sentence is written is somehow confusing, as you only run two models (pre & post-breeding migration), but from the text it seems as you run a lot. As building a RF model implies using different training data subsample, I recommend removing this from the sentences “among the models built with different training data subsamples”.

Authors: In fact, as it was stated in the text, any random forest model is trained using bootstrap aggregation, where each new tree is fit from a bootstrap sample of the training observations (i.e., bagging procedure). Therefore, when calibrating a random forest model, we are actually running multiple iterative trees. Therefore, we can derive errors (and confidence intervals) from the different trees calibrated (see also answer to the comments above). 

Reviewer: L308-310. This result could be due to a temporal bias in sampling effort from observatories. You should show, at least, the number of days of observation per Latitude and Year to support this result.

Fig. 5, figure caption: I think it is more properly as: “Predictions of Balearic shearwater spatial distribution and abundance during migration”

Authors: Since the study period is 13 years (2005-2017) and the study area contains 40,337 spatial cell units at the modelled resolution, it is not feasible to easily show together annual sightings per location and year, neither in a map nor in a table. However, the proper spatial representativeness of the data used in this study is now reported in Figure 1 as graduated symbols representing the number of sightings (see also comments of the editor) per location and migration period (pre-breeding and post-breeding). In addition, sightings per year and month are reported in Figure S1 in the Supplementary Material. 

Regarding the Fig. 5 caption, we have mostly followed the suggestion made by the reviewer#1 and it is now as following: “Predictions of Balearic shearwater spatial distribution of abundance during migration”.

Discussion

Reviewer: L349-352: I understand Big Data is trending topic…But this paper is really far from using Big Data, so I recommend to avoid the use of this term.

Authors: In spite of the relatively reduce data set finally analysed (about thousands of shearwater records), open data sources managed during the analysis and later predictions conducted in our research (mostly NCEP reanalysis -weather predictors) contain more than 1,250,447 rows, thus we think it is worthy of being mentioned as “Big Data”.

Reviewer: L358: My suggestion: “allow to identify general spatio-temporal patterns as data come from different individuals belonging to several populations”.

You should discuss in this part the weakness of data coming from volunteering programmes, such as temporal gaps, uneven sampling effort in space and time, etc. particularly in L353 where you indicate “its characteristic noise” but do not explain anything more.

Authors: We have followed the suggestion made by the reviewer and the text has been changed accordingly. In addition, we have also discussed about the weakness of citizen science data and we have added a few citations to support our discussion.

Reviewer: L361-365: This phrase is too long, 5 lines…. Please rephrase or split it.

Authors: Done.

Reviewer: L368: RF are not good to forecast out of the range of input data, so future projections based on this could be unreliable… depending on input data. Moreover, I think this issue (I mean future projections) is out of the scope of this paper.

Authors: We agree with the reviewer and we have removed any mention to future projections derived from the models.

Reviewer: L371: ref [74] is about ducks. I would say there are more appropriates references to cite here related with seabirds.

Authors: We have followed the suggestion made by the reviewer and the original citation has been replaced by a more suitable one (Tsikliras et al. 2019).

Reviewer: L379: format of references incorrect.

Authors: The citation was misplaced thus we have removed it in the new version of the manuscript.

Reviewer: L382: Be consistent with therm used. Replace volunteer data by scitizen science data.

Authors: Done.

Reviewer: L393: what do you mean to “larger variability observed”? which result support this?

Authors: We referred to the wider confidence intervals in the model accuracy measures. This has now been clarified in the text.

Reviewer: L395-397: This phrase is not clear. Please rephrase it.

Authors: Done.

Reviewer: L422: this is related to some clarifications needed that I mentioned before.

Authors: We meant “when variability in accuracy of the predicticted abundance is larger”. This has now been clarified in the text. See also comment above.

Reviewer: L433-448. I found Conclusions section clear and nice

Authors: Thank you!

Bibliography:

Reviewer: There are typos spread over the section. 

Authors: We have corrected the typos throughout this section.

6. PLOS authors have the option to publish the peer review history of their article (what does this mean?). If published, this will include your full peer review and any attached files.

Do you want your identity to be public for this peer review? For information about this choice, including consent withdrawal, please see our Privacy Policy.

Reviewer #1: No

---

## [Decision Letter · Decision Letter 1]

23 Apr 2020

PONE-D-19-29580R1

Citizen science for predicting spatio-temporal patterns in seabird abundance during migration

PLOS ONE

Dear Dr Martin,

Thank you for submitting your manuscript to PLOS ONE. After careful consideration, we feel that it has merit but does not fully meet PLOS ONE’s publication criteria as it currently stands. Therefore, we invite you to submit a revised version of the manuscript that addresses the points raised during the review process.

We would appreciate receiving your revised manuscript by Jun 07 2020 11:59PM. To enhance the reproducibility of your results, we recommend that if applicable you deposit your laboratory protocols in protocols.io, where a protocol can be assigned its own identifier (DOI) such that it can be cited independently in the future. For instructions see: http://journals.plos.org/plosone/s/submission-guidelines#loc-laboratory-protocols

We look forward to receiving your revised manuscript.

Kind regards,

Vitor Hugo Rodrigues Paiva

Academic Editor

PLOS ONE

Reviewers' comments:

Reviewer's Responses to Questions

**Comments to the Author**

1. If the authors have adequately addressed your comments raised in a previous round of review and you feel that this manuscript is now acceptable for publication, you may indicate that here to bypass the “Comments to the Author” section, enter your conflict of interest statement in the “Confidential to Editor” section, and submit your "Accept" recommendation.

Reviewer #1: All comments have been addressed

Reviewer #2: (No Response)

2. Is the manuscript technically sound, and do the data support the conclusions?

Reviewer #1: Yes

Reviewer #2: Partly

3. Has the statistical analysis been performed appropriately and rigorously? 

Reviewer #1: Yes

Reviewer #2: No

4. Have the authors made all data underlying the findings in their manuscript fully available?

Reviewer #1: Yes

Reviewer #2: No

5. Is the manuscript presented in an intelligible fashion and written in standard English?

Reviewer #1: Yes

Reviewer #2: Yes

6. Review Comments to the Author

Reviewer #1: I congratulate the authors for changes accounted over the manuscript, which is now much easier to follow, readable and understandable than the previous version. In my opinion they resolved successfully most of the comments done in the first submission. Nevertheless I have some comments on this new version. Despite most of them are minor comments, I really do think authors should amend them to improve the manuscript for publication.

My comments are detailed below. Line numbers correspond to those of the version with “Tracking changes” activated.

L36: I suggest "Random Forest regression models"

L54: ecosystemS, “s” in plural.

L65: have been received (remove “been”)

L74: I suggest replace “limit” by other word such as “undermine”, “affect”.

L90: I suggest: species migratory range due to the variety of migratory strategies at colony- and individual levels.

L98: replace “Therefore” by other term to avoid starting in the same way than previous sentence, it would make reading more fluid.

L98-99: I suggest: even providing very detailed information on INDIVIDUAL SEABIRDS’ movements, have a limited ability…

L101: My suggestion: Census methods, in contrast, can provide a VALUABLE overall picture, even though they do not allow to detect birds when they use areas out of human sight.

L106-131. Just another suggestion to ease reading. You used “however” and “in this sense” repeatedly to start sentences. May be you can replace some of them for readability.

L130: as you say “almost unexploited” I would expect to see at least one reference using Trektellen. If there are no publications, it may be better to say “have remained unexploited by the scientific community to the best of our knowledge” or something similar.

L143: Replace “geolocator archival tags” by “geolocation archival tags”, which is the correct form.

L167: may provide a useful records (delete “a”)

L178-179: the correct name does not include “Individual”, just “Species Distribution models”.

L263: How many sighting points from eBird?

L275: Why “Environmental” with capital letter?

L277: I suggest to put a comma before “thus” to ease reading. This advice also applies to many other "thus" along the manuscript.

L288: Caption of Table 1. For consistency you should say “Description of environmental predictors.”.

L393: Classification and Regression TreeS, add “s”.

L414-417: Honestly, to the best of my knowledge there is no need to apply any kind of transformation on the response variable to run Random Forest models. Conversely to GLMM and other parametric methods, random forest do not have any kind of assumption about distribution of data or residuals, so that count data adjust to a Poisson distribution is irrelevant here. The same about the predictive power of the model, it will not change because of transforming the response variable. A different thing is to transform predictors applying standardization, but that is different and is not what was done here. To make sure about my previous knowledge, I did a little research trying to find support to your statement, with no luck. May be I am wrong, so I sincerely encourage the authors to provide a reference supporting that transforming the response variable improves accuracy and increases predictive power of Random Forest models. If you do not find such support, I think the log transformation could be simply justified to improve interpretability of data -which (to me) should be enough-, but not to improve accuracy and predictive power of Random Forest.

L515: Regarding Fig. 4. Please (1) indicate in legend that values are in ln, or back transform them to the actual scale, (2) change year to factor so in the X axis years do not appear with decimal digits.

L532: Caption of Figure 5. I think “spatial distribution of abundance” can be confusing for readers. My suggestion to the entire caption to ease readability: “Predicted abundance of Balearic shearwater across its distribution range during migration. Colour gradient indicates the percentage of the maximum predicted abundance across the study area on the fifteenth of the month, from May to December, from environmental conditions (see Table 1) occurred in year 2017."

L553: “Our results showed (…) successfully applied to describe the migratory distribution (AND ABUNDANCE?) of seabird species accurately.”

L579: increaseD, add "d"

L587-596: “In contrast…(…) in the long-term.” Phrase too long, more than five lines…hard to read. Please split it.

L636-637: “For instance, locations with tailwinds, high abundance level and low chlorophyll concentration”, reorder, as in the next sentence, to: For instance, locations with high abundance level, tailwinds, and low chlorophyll concentration”.

L650: “But the major achievement of our modelling attempt is the high temporal resolution that we achieved with our models”. May be better to rephrase this phrase to avoid tongue twisters….for instance substituting “achievement” by accomplishment, and “modelling attempt” by analysis (in fact, as you already performed the modelling, it is no longer an attempt, isn’t it?).

L676: “the two migration movements”. I suggest to change this by “the two migratory periods” for clarity, and for consistency (see your lines 266-267).

L711: In relation to successful approaches developed to identify marine Important Bird Areas, instead of reference [69] I find much more appropriate and relevant to cite the work developed by SEO/BirdLife, a fortiori, in a work regarding Balearic Shearwater. Either the working-example paper https://www.sciencedirect.com/science/article/abs/pii/S0006320711004745 or the book Arcos, J. M. (2009). Áreas importantes para la conservación de las aves marinas en España: Important areas for the conservation of seabirds in Spain. Sociedad Española de Ornitologia (SEO/BirdLife).

Reviewer #2: First of all, I’d like to congratulate the authors for the study. The study, definitely, is interesting and relevant as it is aiming at (1) a better understanding of the migration pathways of a highly threatened seabird species whose general knowledge present several gaps, and (2) deal with citizen-science large data banks. Such data banks are very useful and studies that aids in a better understanding of how to analytically deal with citizen science in order to provide biologically and ecologically sound information are always welcome. I have no doubt that this study deserves attention and have potential to make an important contribution on those two topics I mentioned above. Therefore, I did a careful reading of the manuscript and checked with attention the critics of the previous reviewers and the authors’ response. I know the reviewing process can be daunting sometimes, particularly when a first round of reviews have already been done, and then a new reviewer comes in and makes further critics that haven’t been there before. Unfortunately, I think that is going to be my role here. In my view, the main fragilities of this study are: the details of the analytical approach need to be better presented; presentation of results needs to be reviewed, several results would be better presented if figures were made differently and/or a different set of statistical outputs of the RF models were used; and, finally the most critical issue, I ask authors to reconsider using Lat and Long as factors in the modelling, based on figure 3, both Lat and Long are the variables that explained most of models’ power of prediction – what is the consequence of that to the findings? If authors think the use of LatLong is justified, they should explain it carefully in methods and discuss it more deeply. But for me, it is clear on Fig3 that LatLong reduced considerably the importance of the environmental variables and the model is mostly predicting spatial occurrences than habitat use. Below a list of the detailed comments that authors should address.

I ask patience for the authors and hope they understand the comments and critics aims for improving the final manuscript. Have a good review, looking forward to read the reviewed version!

L52 – Seabirds instead of “marine birds”

L54 – I suggest deleting “migrate” from this sentence. Start next sentence with something like “migratory seabirds are particularly susceptible to a large number of stressors given the variety of habitats they use throughout their year-cycle…”

L60- Suggest rephrasing to “Stopovers are key sites; conditions experienced by seabirds in stopover can affect individual survival throughout migration and drive population dynamics” or some similar idea.

L76 – “and in relation to specific individual traits” this is a bit “loose” within the sentence. I suggest you describe which are those individual traits, or delete this part of the sentence. For instance, in the start of the sentence “This is particularly true for long-lived seabirds with a defined set of traits such as….. which have a great capacity… “

L96 – Rephrase “Prominent electronic citizen science data banks includes eBird and trekellen xxxx, two biodiversity…”. Additionaly, why they are prominent? Cited studies compared them with other databanks? Maybe more popular, commonly used, or else…?

L104 - Birdlife factsheet indicates several parameters explaining why this shearwater is critically endangered and why actions to improve knowledge on population trends are urgent, but I did not find any reference to the species being one of the most threatened species of seabirds. Maybe rephrase emphasizing that information from [26] allows placing this species as one of the most threatened seabirds in the world, as possibly it is.

L151. Is it possible to plot the position of known breeding colonies of the species?

Line 170. This is key: correctly identified when sighted. Some seabirds at sea are very difficult to distinguish; therefore the use of data from citizen science for study difficult-to-identify species should be careful. I hope that it is acknowledged in the discussion.

L183. Data from 2005 to 2017 was used because it is the most representative period in terms of sampling. 2018 was excluded because it does not have a homogeneous sampling throughout the year. Mention it here.

L191. It seems you used year and geographical position as factors in your model (by seeing figure 3). It is not described in methods how those variables entered the model. I would not recommend using Lat and Long as factors in this case, and I really think it doesn’t help your study at all (if it does, please provide an explanation). Lat and Long are not environmental variables used by the birds, and given that both were the most influencing variables in the model, it is likely the models outputs are predicting the occurrences rather than the relation between abundance and environment. Don’t you agree? If this is true, the model should be run again without lat and long as factors. You could, a posteriori, use a probability of occurrence based on latlong to filter the predictions like Hindell et al. (2020; DOI 10.1038/s41586-020-2126-y) did.

L193. In practical terms, zero was suppressed, right?

L197 – “For a detailed description of the set of predictors included in the models, please see the Supplementary Material.” Instead of this, you could add in line 195 “…predictors (Table 1, Supp. Material).” The references in the supplemental material are the same from the main text? Shouldn’t sup. material have its own reference list?

L211. Where exactly in results? Reference to supp. material.

L212-221. I didn’t understand this section. You applied a single machine-learning based technique (random forest), right? You start the sentence saying it in plural. Then you presented a series of models and an equation, apparently presented in three studies [43,44,45]. If this is true, this paragraph could be simplified to saying that among a handful of techniques, RF was identified as one of the most accurate in at least three studies based on comparison of RMSE. The reader can check then the studies that you cited.

L242 – 248.

This is not clear and needs further explaining. You used bootstrap to calibrate the model, the algorithm stopped when the best “solution” was reached? That’s why you have a variable number of combinations (1-15)? It is also usual to run a fixed number of trees and check the increase in accuracy, and a posteriori, select the trees that produced increasing accuracy without substantial overfit. That’s why you had a variable number of trees between 100 to 500? I imagine you used 500 threes for all the possible variable combinations and used the approach I described to select the best solutions, the minimum number of trees required to achieve that was 100. Is it right? Or I completely misunderstood?

The RMSE is calculated over the OOB-error? It did not seem that you did it. In my opinion, a plot showing OOB vs RMSE (or AUC, or other…) over all iteration trees could provide such information and justify why there is a variable number of trees used in the final and averaged abundance output (I briefly checked the packages you used, there are some ways of doing that: http://topepo.github.io/caret/model-training-and-tuning.html). If this is not what you did and I misunderstood, please provide alternative explanations.

L272. Order of the figures was quite confusing… figure 1 was the last in the PDF… putting the legends with the figures instead of merging it in the text would facilitate reviewing. I hope authors consider it in further revisions or submissions.

L273. You said in methods that you removed correlated variables, and in the end, you used all variables because no variable was correlated. Change it in methods.

L277. Figure 2 points out a small variability on power of prediction based on R2 values, more or less between 0.49 and 0.53, and slightly more accurate predictions during RFPost, as mean error was lower. It is not clear to me how it indicated substantial variation among models. Please explain. Further information is also required: are those results from all the iterated trees?

L279-281. Please explain what lagged-iterated differences mean and how they were calculated. Is this a lag between iterations or annual variability? Check next comment.

L299. Year entered as a factor? Or models were run separately for each year? Methods were not clear on how annual variability was used in the models. If year entered as factor in the analysis, how did you deal with fisheries time-coverage being different then the species data? You probably used fishing as a fixed non-dynamic variable. How you justify that?

L306. What is minimum depth? It is crucial to understand figure 3. An alternative way (more straightforward, in my opinion) of analyzing variable contribution is to plot the change in accuracy when the variable is absent from the model. Seems that Caret package has a standard function to do that: ‘varImp’.

L312. Not sure whether figure 4 contributes to the overall results. It could be placed on the suppl. Material. How annual variability was used in the model is not clear either.

L325. You said in methods you used two periods of migration, instead you present here eight different periods. Can you group information for only those two period? More detailed results could be placed in the Supp Material and in the main text a general figure highlighting the detected stop-overs on the two different periods. Breeding and non-breeding known areas in the figure also would be very useful.

L325. Figure suggests part of the population remains in the Mediterranean year-round. Is that true or it is a product of the modelling?

L348-351. It is important to highlight that boosted regression trees methods such as the used here can artificially inflate accuracy with increasing iterations, therefore the need to evaluate how fit and accuracy varies with iterations. It is possible to have increasing accuracy and loss of fit to the point that the model starts predicting the response itself (occurrence or abundance) disregarding the factors, therefore one should use this in order to select the optimum number of iterations to be used in the final model outputs. This is not clear in methods, and this is not discussed either.

L386-390. Nowhere in your results there are estimated response curves. So please cite a reference to this statement.

L389. It would be very useful to place in the map the sites you mention here, such as Alboran Sea or Gulf of Cadiz in figure 1, so readers not familiar with this region of the globe can be spatially situated.

L392-394. I thought results indicated differences on migratory routes between periods. Did I misunderstand?

L397. Yet, how temporal variability was used in the model is not clear.

L417. Again, without estimated response curves, I don’t think it is possible to reach such conclusion. A variable having high importance in the modelling doesn’t mean the birds had higher abundance in the higher values of the variable, this is particularly true using a model as RF that does not necessarily assumes linearity.

L423. How variability in accuracy (that was not as large as the authors claimed) was led by the link with food availability? It is not clear. Needs better explanation.

7. PLOS authors have the option to publish the peer review history of their article (what does this mean?). If published, this will include your full peer review and any attached files.

Reviewer #1: No

Reviewer #2: Yes: Lucas Krüger

---

## [Author Response · Author response to Decision Letter 1]

1 Jun 2020

1st June 2020

PONE-D-19-29580R1

Citizen science for predicting spatio-temporal patterns in seabird abundance during migration

Vitor Hugo Rodrigues Paiva

Academic Editor

PLOS ONE

Dear Mr. Rodrigues,

We sincerely thank you for allowing us to resubmit our paper. Following your instructions, we resubmit the manuscript titled “Citizen science for predicting spatio-temporal patterns in seabird abundance during migration” (PONE-D-19-29580R1), after fully addressing all the concerns raised by the reviewers. 

Changes done throughout the text have been recorded, via track-changes, in a document enclosed with this re-submission. Please find below the detailed explanation of the modifications we have made in response to the concerns raised. A copy of the document with the track changes feature turned off is also attached. Lines indicated below referred to the document with the track changes feature turned on. 

We hope that you will find this revised version suitable for publication in PLOS ONE.

Yours faithfully,

Beatriz Martín on behalf of all the authors

PONE-D-19-29580R1

Citizen science for predicting spatio-temporal patterns in seabird abundance during migration

PLOS ONE

Dear Dr Martin,

Thank you for submitting your manuscript to PLOS ONE. After careful consideration, we feel that it has merit but does not fully meet PLOS ONE’s publication criteria as it currently stands. Therefore, we invite you to submit a revised version of the manuscript that addresses the points raised during the review process.

We would appreciate receiving your revised manuscript by Jun 07 2020 11:59PM. To enhance the reproducibility of your results, we recommend that if applicable you deposit your laboratory protocols in protocols.io, where a protocol can be assigned its own identifier (DOI) such that it can be cited independently in the future. For instructions see: http://journals.plos.org/plosone/s/submission-guidelines#loc-laboratory-protocols

• A rebuttal letter that responds to each point raised by the academic editor and reviewer(s). This letter should be uploaded as separate file and labeled 'Response to Reviewers'.

• A marked-up copy of your manuscript that highlights changes made to the original version. This file should be uploaded as separate file and labeled 'Revised Manuscript with Track Changes'.

• An unmarked version of your revised paper without tracked changes. This file should be uploaded as separate file and labeled 'Manuscript'.

We look forward to receiving your revised manuscript.

Kind regards,

Vitor Hugo Rodrigues Paiva

Academic Editor

PLOS ONE

Reviewers' comments:

Reviewer's Responses to Questions

Comments to the Author

1. If the authors have adequately addressed your comments raised in a previous round of review and you feel that this manuscript is now acceptable for publication, you may indicate that here to bypass the “Comments to the Author” section, enter your conflict of interest statement in the “Confidential to Editor” section, and submit your "Accept" recommendation.

Reviewer #1: All comments have been addressed

Reviewer #2: (No Response)

2. Is the manuscript technically sound, and do the data support the conclusions?

Reviewer #1: Yes

Reviewer #2: Partly

3. Has the statistical analysis been performed appropriately and rigorously? 

Reviewer #1: Yes

Reviewer #2: No

4. Have the authors made all data underlying the findings in their manuscript fully available?

Reviewer #1: Yes

Reviewer #2: No

Authors: All the environmental predictors used in the present study are freely available from the websites of the different data sources indicated in Table 1. Moreover, the abundance of shearwater analysed in this study can be obtained after request from eBird (https://ebird.org/). This information is now explicitly indicated in the Acknowledgements section.

5. Is the manuscript presented in an intelligible fashion and written in standard English?

Reviewer #1: Yes

Reviewer #2: Yes

6. Review Comments to the Author

Reviewer #1: I congratulate the authors for changes accounted over the manuscript, which is now much easier to follow, readable and understandable than the previous version. In my opinion they resolved successfully most of the comments done in the first submission. Nevertheless I have some comments on this new version. Despite most of them are minor comments, I really do think authors should amend them to improve the manuscript for publication.

My comments are detailed below. Line numbers correspond to those of the version with “Tracking changes” activated.

L36: I suggest "Random Forest regression models"

Authors: Done.

L54: ecosystemS, “s” in plural.

Authors: Done.

L65: have been received (remove “been”)

Authors: Done.

L74: I suggest replace “limit” by other word such as “undermine”, “affect”.

Authors: Done.

L90: I suggest: species migratory range due to the variety of migratory strategies at colony- and individual levels.

Authors: Done.

L98: replace “Therefore” by other term to avoid starting in the same way than previous sentence, it would make reading more fluid.

Authors: Done.

L98-99: I suggest: even providing very detailed information on INDIVIDUAL SEABIRDS’ movements, have a limited ability…

Authors: Done.

L101: My suggestion: Census methods, in contrast, can provide a VALUABLE overall picture, even though they do not allow to detect birds when they use areas out of human sight.

Authors: This was a correction made by other reviewer in the previous reviewing round. What we mean here is that census data do not allow to identify the colony or the age of the individuals. We have slightly changed the sentence in order to clarify it for the reader. Please see line 82.

L106-131. Just another suggestion to ease reading. You used “however” and “in this sense” repeatedly to start sentences. May be you can replace some of them for readability.

Authors: Done.

L130: as you say “almost unexploited” I would expect to see at least one reference using Trektellen. If there are no publications, it may be better to say “have remained unexploited by the scientific community to the best of our knowledge” or something similar.

Authors: Done.

L143: Replace “geolocator archival tags” by “geolocation archival tags”, which is the correct form.

Authors: Done.

L167: may provide a useful records (delete “a”)

Authors: Done.

L178-179: the correct name does not include “Individual”, just “Species Distribution models”.

Authors: Done.

L263: How many sighting points from eBird?

Authors: Sighting points are not fixed in eBird observations so the number of observations is the number of sighting points. We have now clarified this point (see line 183).

L275: Why “Environmental” with capital letter?

Authors: Done.

L277: I suggest to put a comma before “thus” to ease reading. This advice also applies to many other "thus" along the manuscript.

Authors: Done. We have changed this throughout the manuscript.

L288: Caption of Table 1. For consistency you should say “Description of environmental predictors.”.

Authors: Done.

L393: Classification and Regression TreeS, add “s”.

Authors: Done.

L414-417: Honestly, to the best of my knowledge there is no need to apply any kind of transformation on the response variable to run Random Forest models. Conversely to GLMM and other parametric methods, random forest do not have any kind of assumption about distribution of data or residuals, so that count data adjust to a Poisson distribution is irrelevant here. The same about the predictive power of the model, it will not change because of transforming the response variable. A different thing is to transform predictors applying standardization, but that is different and is not what was done here. To make sure about my previous knowledge, I did a little research trying to find support to your statement, with no luck. May be I am wrong, so I sincerely encourage the authors to provide a reference supporting that transforming the response variable improves accuracy and increases predictive power of Random Forest models. If you do not find such support, I think the log transformation could be simply justified to improve interpretability of data -which (to me) should be enough-, but not to improve accuracy and predictive power of Random Forest.

Authors: The reviewer is right in relation to the lack of assumptions in the distribution of the data in Random Forest models. However, when we use an approach based on decision trees such as Random forest where data partitioning is applied, we can obtain better results if we work with a dependent variable homogenously distributed, because the dispersion increases as long as the variable increases. We cannot provide a reference but we have provided our own results (see lines 255-263). 

L515: Regarding Fig. 4. Please (1) indicate in legend that values are in ln, or back transform them to the actual scale, (2) change year to factor so in the X axis years do not appear with decimal digits.

Authors: Done. We have also clarified the scale of the reported predictions in the legend of Figure 5 and Figures S4-S6 in the Supplementary material.

L532: Caption of Figure 5. I think “spatial distribution of abundance” can be confusing for readers. My suggestion to the entire caption to ease readability: “Predicted abundance of Balearic shearwater across its distribution range during migration. Colour gradient indicates the percentage of the maximum predicted abundance across the study area on the fifteenth of the month, from May to December, from environmental conditions (see Table 1) occurred in year 2017."

Authors: We have accepted the suggestion made by the reviewer. Please see the new Figure legend.

L553: “Our results showed (…) successfully applied to describe the migratory distribution (AND ABUNDANCE?) of seabird species accurately.”

Authors: Done.

L579: increaseD, add "d"

Authors: Done.

L587-596: “In contrast…(…) in the long-term.” Phrase too long, more than five lines…hard to read. Please split it.

Authors: Done.

L636-637: “For instance, locations with tailwinds, high abundance level and low chlorophyll concentration”, reorder, as in the next sentence, to: For instance, locations with high abundance level, tailwinds, and low chlorophyll concentration”.

Authors: Done.

L650: “But the major achievement of our modelling attempt is the high temporal resolution that we achieved with our models”. May be better to rephrase this phrase to avoid tongue twisters….for instance substituting “achievement” by accomplishment, and “modelling attempt” by analysis (in fact, as you already performed the modelling, it is no longer an attempt, isn’t it?).

Authors: Done.

L676: “the two migration movements”. I suggest to change this by “the two migratory periods” for clarity, and for consistency (see your lines 266-267).

Authors: Done.

L711: In relation to successful approaches developed to identify marine Important Bird Areas, instead of reference [69] I find much more appropriate and relevant to cite the work developed by SEO/BirdLife, a fortiori, in a work regarding Balearic Shearwater. Either the working-example paper https://www.sciencedirect.com/science/article/abs/pii/S0006320711004745 or the book Arcos, J. M. (2009). Áreas importantes para la conservación de las aves marinas en España: Important areas for the conservation of seabirds in Spain. Sociedad Española de Ornitologia (SEO/BirdLife).

Authors: Done.

Reviewer #2: First of all, I’d like to congratulate the authors for the study. The study, definitely, is interesting and relevant as it is aiming at (1) a better understanding of the migration pathways of a highly threatened seabird species whose general knowledge present several gaps, and (2) deal with citizen-science large data banks. Such data banks are very useful and studies that aids in a better understanding of how to analytically deal with citizen science in order to provide biologically and ecologically sound information are always welcome. I have no doubt that this study deserves attention and have potential to make an important contribution on those two topics I mentioned above. Therefore, I did a careful reading of the manuscript and checked with attention the critics of the previous reviewers and the authors’ response. I know the reviewing process can be daunting sometimes, particularly when a first round of reviews have already been done, and then a new reviewer comes in and makes further critics that haven’t been there before. Unfortunately, I think that is going to be my role here. In my view, the main fragilities of this study are: the details of the analytical approach need to be better presented; presentation of results needs to be reviewed, several results would be better presented if figures were made differently and/or a different set of statistical outputs of the RF models were used; and, finally the most critical issue, I ask authors to reconsider using Lat and Long as factors in the modelling, based on figure 3, both Lat and Long are the variables that explained most of models’ power of prediction – what is the consequence of that to the findings? If authors think the use of LatLong is justified, they should explain it carefully in methods and discuss it more deeply. But for me, it is clear on Fig3 that LatLong reduced considerably the importance of the environmental variables and the model is mostly predicting spatial occurrences than habitat use. Below a list of the detailed comments that authors should address.

I ask patience for the authors and hope they understand the comments and critics aims for improving the final manuscript. Have a good review, looking forward to read the reviewed version!

Authors: We have followed the suggestions and the recommendations made by the reviewer in relation to the Methods and Results presented in the study, please see comments below. Regarding the longitude and latitude as predictors because, we must have in mind that we are not predicting the static distribution of the shearwaters as a particular time snapshot, but their occurrence along the migration period. Apart from environmental cues such as food availability, migratory birds can more or less rely on photoperiodic cues and/or endogenous rhythms to initiate their migration. Longitude, latitude and date variables allow us to include in the models the endogenous rhythm of the bird. As we can see from the variable importance in the models, whereas food availability seems to be a more important triger of post-breeding migration, the internal rhythm of the bird appears to be more important during the pre-breeding period. For further details, please see comment below and new lines 210-228.

L52 – Seabirds instead of “marine birds”

Authors: Done.

L54 – I suggest deleting “migrate” from this sentence. Start next sentence with something like “migratory seabirds are particularly susceptible to a large number of stressors given the variety of habitats they use throughout their year-cycle…”

Authors: Done.

L60- Suggest rephrasing to “Stopovers are key sites; conditions experienced by seabirds in stopover can affect individual survival throughout migration and drive population dynamics” or some similar idea.

Authors: We have mostly followed the reviewer suggestion. However, since conditions in stopovers may affect survival beyond the migratory journey due to carry-over effects, we have slightly changed the re-phrasing (see lines 61-64). See also the correction made by the reviewer #1 above.

L76 – “and in relation to specific individual traits” this is a bit “loose” within the sentence. I suggest you describe which are those individual traits, or delete this part of the sentence. For instance, in the start of the sentence “This is particularly true for long-lived seabirds with a defined set of traits such as….. which have a great capacity… “

Authors: By “traits” we meant sex, age, breeding colony. This has now explicitly written in the text.

L96 – Rephrase “Prominent electronic citizen science data banks includes eBird and trekellen xxxx, two biodiversity…”. Additionaly, why they are prominent? Cited studies compared them with other databanks? Maybe more popular, commonly used, or else…?

Authors: They are prominent in number of users. We have clarified this point in the text.

L104 - Birdlife factsheet indicates several parameters explaining why this shearwater is critically endangered and why actions to improve knowledge on population trends are urgent, but I did not find any reference to the species being one of the most threatened species of seabirds. Maybe rephrase emphasizing that information from [26] allows placing this species as one of the most threatened seabirds in the world, as possibly it is.

Authors: Done.

L151. Is it possible to plot the position of known breeding colonies of the species?

Authors: We think that it is not a good idea to show the breeding colonies in the map because only a few main colonies are known for these species and the information would not be an overall picture of the breeding range. However, following the reviewer suggestion and according to the publication below, we now show both the breeding and the non-breeding range for the species in Figure 1.

José Manuel Arcos: International species action plan for the Balearic shearwater, Puffinus mauretanicus, SEO/BirdLife & BirdLife International, 2011, 

Line 170. This is key: correctly identified when sighted. Some seabirds at sea are very difficult to distinguish; therefore the use of data from citizen science for study difficult-to-identify species should be careful. I hope that it is acknowledged in the discussion.

Authors: Actually, it was mentioned in the Discussion section (see lines 426-431; and line 516). 

L183. Data from 2005 to 2017 was used because it is the most representative period in terms of sampling. 2018 was excluded because it does not have a homogeneous sampling throughout the year. Mention it here.

Authors: In fact this point had been indicated just in a couple lines below in the previous version of the manuscript: “However, to ensure the maximum seasonal representation of the dataset to be analysed (see Figure S1 in the Supplementary material), from this total sample of birds, we only modelled data collected during the migration period from 2005 to 2017 (see Results)”. Please see lines 188-191.

L191. It seems you used year and geographical position as factors in your model (by seeing figure 3). It is not described in methods how those variables entered the model. I would not recommend using Lat and Long as factors in this case, and I really think it doesn’t help your study at all (if it does, please provide an explanation). Lat and Long are not environmental variables used by the birds, and given that both were the most influencing variables in the model, it is likely the models outputs are predicting the occurrences rather than the relation between abundance and environment. Don’t you agree? If this is true, the model should be run again without lat and long as factors. You could, a posteriori, use a probability of occurrence based on latlong to filter the predictions like Hindell et al. (2020; DOI 10.1038/s41586-020-2126-y) did.

Authors: Latitude and longitude were included as discrete variables in the models (this is now explicitly indicated in line 212). We have included latitude and longitude as predictors because we are not predicting the static distribution of the shearwaters as a particular time snapshot, but their occurrence along the migration period. Below we provide a detailed explanation. Depending on the species, birds can more or less rely on photoperiodic cues and/or endogenous rhythms to initiate their migration. Therefore, although migration decisions in shearwaters are partially dictated by food availability, the endogenous programme of the bird should make it to instinctively move into the north as long as the post-breeding season progresses. Date, longitude and latitude variables allow us to include in the models the endogenous rhythm of the bird. In addition, as we mentioned in the Introduction section, breeding and wintering grounds of different shearwater populations differ. Therefore, longitude and also latitude can be indirect proxies of the effects that different wintering sites, different length of the route and different en-route environmental conditions may pose to different migrant birds. These effects are not possible to be quantified in detail from citizen datasets since, in contrast to ringing or electronic tracking devices records, because they do not provide information on the particular origin and destination of the observed bird. On the other hand, and most important, we cannot ignore that we are not predicting merely habitat use but modelling birds which are in movement. In this sense, the use of these additional predictors also allows to incorporate the effects of the complex interactions emerging between predictors, such as latitude, date and temperature. For instance, for a migrating shearwater it has not the same meaning, in terms of migration decisions, to experience 25°C in the Mediterranean Sea in mid-February than 25°C in the UK coasts in mid-August. Similarly, due to different rates of change across space among different bird populations, interactions between predictors and “latitude”, and between “year” and “latitude” may allow to quantify both the spatial and temporal heterogeneity in the migratory responses. We have now briefly clarified all these facts in the new version of the manuscript (please see lines 212-226).

L193. In practical terms, zero was suppressed, right?

Authors: The reviewer is right. We have now clarified this point in the text (please see line 199).

L197 – “For a detailed description of the set of predictors included in the models, please see the Supplementary Material.” Instead of this, you could add in line 195 “…predictors (Table 1, Supp. Material).” The references in the supplemental material are the same from the main text? Shouldn’t sup. material have its own reference list?

Authors: Done. The sentence has been removed and we have created a new specific section of references in the Supplementary material.

L211. Where exactly in results? Reference to supp. material.

Authors: Done.

L212-221. I didn’t understand this section. You applied a single machine-learning based technique (random forest), right? You start the sentence saying it in plural. Then you presented a series of models and an equation, apparently presented in three studies [43,44,45]. If this is true, this paragraph could be simplified to saying that among a handful of techniques, RF was identified as one of the most accurate in at least three studies based on comparison of RMSE. The reader can check then the studies that you cited.

Authors: Actually, as the reviewer suggests, this assessment was only cited in the first version of the manuscript. This paragraph was added in the previous reviewing round after request of one of the reviewers, since the cited work remains yet unpublished (it is under review). For this reason, we have decided to keep the further explanation regarding the selection of the modelling approach.

L242 – 248.

This is not clear and needs further explaining. You used bootstrap to calibrate the model, the algorithm stopped when the best “solution” was reached? That’s why you have a variable number of combinations (1-15)? It is also usual to run a fixed number of trees and check the increase in accuracy, and a posteriori, select the trees that produced increasing accuracy without substantial overfit. That’s why you had a variable number of trees between 100 to 500? I imagine you used 500 threes for all the possible variable combinations and used the approach I described to select the best solutions, the minimum number of trees required to achieve that was 100. Is it right? Or I completely misunderstood?

The RMSE is calculated over the OOB-error? It did not seem that you did it. In my opinion, a plot showing OOB vs RMSE (or AUC, or other…) over all iteration trees could provide such information and justify why there is a variable number of trees used in the final and averaged abundance output (I briefly checked the packages you used, there are some ways of doing that: http://topepo.github.io/caret/model-training-and-tuning.html). If this is not what you did and I misunderstood, please provide alternative explanations.

Authors: We think the reviewer confuse the machine learning technique applied. Bagging is the application of the Bootstrap procedure to a high-variance machine learning algorithm, typically decision trees. And Random Forest is an improvement over bagged decision trees, consisting of a large number of individual decision trees that operate as an ensemble. As it is stated in the text (see lines 267-269), any random forest model applies bagging (i.e., bootstrap aggregating) to sub-sample the data that are used for training. Please, see also comment below about the difference between boosting and bagging techniques. 

As it is described in the Methods, we are following a usual Random Forest regression analysis so the RMSE is calculated over all the trees in a particular model over the bagging procedure. In Random forest each model is calculated through a bagging procedure (i.e., by building multiple different decision tree models from a single training data set by repeatedly using multiple bootstrapped subsets of the data and averaging the models, see lines 267-269). OOB error is derived from this bootstrap process. When bagging decision trees the number of samples and hence the number of trees is a parameter to be selected (see lines 278-279). Therefore, the algorithm increases the number of trees on run after run until the accuracy begins to stop showing improvement. We have now clarified this point in the text (please see lines 278-282). On the other hand, random forest model has hyperparameters that should be tuned to avoid overfitting. The previous is now explicitly clarified in the text (lines 273-274). Therefore, as it was stated in the text, to determine the parameter values offering the best fit, we specified a set of tuning values to be tested during the calibration of the models. That is why we have a number of combinations (1-15). Specifically, we applied a grid search method, thus we evaluated the model over different combinations of parameters included in the grid (values ranging between 1-15 at one-unit intervals). We must have in mind that RMSE is the average difference between the observed known values of the outcome and the predicted value by the model (see lines 243-244). Therefore, the best fit parameter combination is then asseses by means of the RMSE of the models (see new lines 282-284 explicitly explaining the later point). 

We could not apply AUC (Area Under the Receiver Operating Characteristics -ROC- Curve) because this measure is only applicable to classification trees but not to regression trees, as it is our approach. That is the reason why we chose RMSE metric. 

L272. Order of the figures was quite confusing… figure 1 was the last in the PDF… putting the legends with the figures instead of merging it in the text would facilitate reviewing. I hope authors consider it in further revisions or submissions.

Authors: we will take care of the figure ordering when re-submitting the manuscript. However, the inclusion of the legends throughout of the text is a requirement of PLOSONE to indicate the suggested position of the Figure within the text. 

L273. You said in methods that you removed correlated variables, and in the end, you used all variables because no variable was correlated. Change it in methods.

Authors: Done, see lines 232-235.

L277. Figure 2 points out a small variability on power of prediction based on R2 values, more or less between 0.49 and 0.53, and slightly more accurate predictions during RFPost, as mean error was lower. It is not clear to me how it indicated substantial variation among models. Please explain. Further information is also required: are those results from all the iterated trees?

Authors: The differences between pre-breeding and post-breeding models were measured as the lagged and iterated differences over the model resamples. This sentence was initially included in the Result section, but we have now moved it into the Methods. On the other hand, as it is stated in the text, the variation among models built with different training data subsamples is measured as the 95% confidence interval in the used metrics (RMSE and R2). We have now qualified this point in the text. Even if the variability seems small, according to the Results (see lines 328-338), differences in RMSE between pre- and post-breeding models are statistically significant.

L279-281. Please explain what lagged-iterated differences mean and how they were calculated. Is this a lag between iterations or annual variability? Check next comment.

Authors: As it is stated in the text, “the differences between pre-breeding and post-breeding models were measured as the lagged and iterated differences over the model resamples”, so the differences are measured over the model resamples, as the difference between a resample and the previous one in an iterative way. We have now included this clarification in the text (lines 328-329).

L299. Year entered as a factor? Or models were run separately for each year? Methods were not clear on how annual variability was used in the models. If year entered as factor in the analysis, how did you deal with fisheries time-coverage being different then the species data? You probably used fishing as a fixed non-dynamic variable. How you justify that?

Authors: Year, as well as date, longitude and latitude, entered in the models as discrete variables predicting shearwater abundance. We have now clarified this point (see comment above and lines 210-2132). 

 L306. What is minimum depth? It is crucial to understand figure 3. An alternative way (more straightforward, in my opinion) of analyzing variable contribution is to plot the change in accuracy when the variable is absent from the model. Seems that Caret package has a standard function to do that: ‘varImp’.

Authors: Minimal depth is a method developed by Ishwaran et al. (2010) that determines variable importance by the position of the variables in the decision trees so the importance is based on the decision tree structure. The idea is that variables that tend to split close to the root node should have more importance in prediction. We really think that our approach, even more complex than other simple measures of prediction performance, provides a better understanding of the relative variable importance and, therefore, it should be kept. To help the reader understanding the approach, we have included additional information on the way that minimal depth is measured (see lines 298-302).

L312. Not sure whether figure 4 contributes to the overall results. It could be placed on the suppl. Material. How annual variability was used in the model is not clear either.

Authors: We disagree with the reviewer regarding the importance of Figure 4. The results on the interaction between latitude and year variables (which is shown in Figure 4) allow to determine whether there is a spatial trend in abundance over the period. This is explained in lines 340-342. This fact has been previously discussed for this and other seabird species in other research studies, see the Discussion section, and it is relevant both for understanding the effects of climate change and to delimit marine areas (see the Discussion section). For these reasons, we have kept Figure 4 within the main text of the article. Regarding annual variability, as it was stated in the text (see line 213), interannual variability is measured through the effect of the variable “year”. If there is no interannual variability, then the “year” should have a minor importance in the model. We have further clarified this fact in the new version of the manuscript (see line 346-349).

L325. You said in methods you used two periods of migration, instead you present here eight different periods. Can you group information for only those two period? More detailed results could be placed in the Supp Material and in the main text a general figure highlighting the detected stop-overs on the two different periods. Breeding and non-breeding known areas in the figure also would be very useful.

Authors: As it is stated in the Methods section (lines 190-196), actually we use two periods of migration (pre-breeding and post-breeding) so we built two specific models for one each other. However, also in the Methods section (lines 305-307), we indicate that “as an example of the potential marine areas that can be identified from the models, we predicted shearwater abundance across the study area on the fifteenth of the month, from May to December, based on the environmental conditions during 2017”, which are the eight different periods mentioned by the reviewer. As it is state in the text (lines 370-374), May to August represent the northward migration whereas September to December show the southward migration period. However, we have now explicitly indicated (see line 374 and new additions in the legend of the Figure 5) that May-August correspond to the post-breeding period and September-December to the pre-breeding. Therefore, we think that the information for both periods in Figure 5 is clearly differentiated.

Regarding the stop-overs, as it is highlighted in the Discussion section (see lines 436-437) , “our predictions should be taken rather as an illustration than as an exact calculation, since they are based on a single snapshot on the fifteenth of the month in a particular year”. An accurate estimation of stop-overs along the migratory route requires to quantify the distribution of abundance along the period in both seasons and apply appropriate methods for delimiting the marine areas (such as spatial prioritization algorithms). This work is part of the present research of the co-authors of the manuscript.

Regarding breeding and non-breeding areas, we think that adding these layers into Figure 5 would make hard to interpret the model results. However, this information has now been added in the new Figure 1.

L325. Figure suggests part of the population remains in the Mediterranean year-round. Is that true or it is a product of the modelling?

Authors: A variable fraction of the total Balearic shearwater population, mainly adult birds, appeared to remain in the Mediterranean (Ruiz and Martí, 2004). This is now discussed in the manuscript (see lines 497-500).

Ruiz, A., Martí, R., 2004. La Pardela Balear. SEO/BirdLife-Conselleria de Medi Ambient del Govern de les Illes Balears, Madrid, Spain.

L348-351. It is important to highlight that boosted regression trees methods such as the used here can artificially inflate accuracy with increasing iterations, therefore the need to evaluate how fit and accuracy varies with iterations. It is possible to have increasing accuracy and loss of fit to the point that the model starts predicting the response itself (occurrence or abundance) disregarding the factors, therefore one should use this in order to select the optimum number of iterations to be used in the final model outputs. This is not clear in methods, and this is not discussed either.

Authors: We think that the reviewer confuses “boosting” with “bagging” (used in Random Forest, see also comment above). As it is clearly stated in the manuscript (see lines ¿?), we do not apply boosted regression trees but Random Forest models. And any random forest model apply bagging (i.e., bootstrap aggregating) to sub-sample the data that are used for training. Random forests builds each tree independently while boosting builds one tree at a time and, while random forests combine results at the end of the process (by averaging or "majority rules"), boosting combines results along the way. For these reasons, boosting can result in better performance than random forests. However, in situations where a lot of noise is present (as it is our case), boosting may not be a good choice because it can result in overfitting (as the reviewer pointed out). Furthermore, Random Forest is an ensemble of decision trees. Although the single decision tree is very sensitive to data variations and it can easily overfit to noise in the data, when we add trees to the Random Forest then the tendency to overfitting decreases thanks to bagging and random feature selection. For all the previous, we found Random Forest as the best modelling approach for our datasets (see also Methods section on the assessment of other modelling techniques). 

L386-390. Nowhere in your results there are estimated response curves. So please cite a reference to this statement.

Authors: See new Figures S4-S6 in the Supplementary material and line 442 in the Discussion section.

L389. It would be very useful to place in the map the sites you mention here, such as Alboran Sea or Gulf of Cadiz in figure 1, so readers not familiar with this region of the globe can be spatially situated.

Authors: Done.

L392-394. I thought results indicated differences on migratory routes between periods. Did I misunderstand?

Authors: In the Results section it is stated that, “according to the models, Balearic Shearwaters use slightly different regions during their northward (May–August, post-breeding) and southward migration (September–December, pre-breeding)”. We think that this is indeed not in contradiction with the above-mentioned sentence by the reviewer (“our results also suggested that Atlantic coast of Portugal and France are migratory, stopover and/or moulting sites for Balearic shearwaters both during pre and post-breeding migration”. As the regions are slightly different, Atlantic areas in Portugal and France are both used during pre- and post-breeding whereas other areas (also detailed in the Discussion section a few lines below) are clearly different for both migration periods (see lines 451-456: “the post-breeding model showed that Western English Channel is an important area for Balearic shearwaters mainly during August, whereas high abundance of shearwaters is predicted along the Algerian/Tunisian coastline in November”).

L397. Yet, how temporal variability was used in the model is not clear.

Authors: Temporal variability is explicitly modelled by “date” and “year” predictors, as well as, of course, indirectly modelled by the temporal variability in the environmental predictors. Please see lines 212-214 in the Methods section and also the Supplementary Methods in the Supplementary material describing the environmental variables modulating the migratory behaviour of shearwaters. See also comments above regarding the “effect to year” and the interaction between year and latitude in Figure 4.

L417. Again, without estimated response curves, I don’t think it is possible to reach such conclusion. A variable having high importance in the modelling doesn’t mean the birds had higher abundance in the higher values of the variable, this is particularly true using a model as RF that does not necessarily assumes linearity.

Authors: We have now added additional information in this regard in the Supplementary Material. Please see new Figures S4-S6.

L423. How variability in accuracy (that was not as large as the authors claimed) was led by the link with food availability? It is not clear. Needs better explanation.

Authors: As it is stated in the Results section, the difference in RMSE (i.e. model accuracy) between pre- and post-breeding models, even though small, is statistically significant at a Bonferroni p-level (see new addition in line 330 to highlight this point). If the distribution of the shearwater abundance depends on static predictors such as bathymetry, then the differences among resamples built with different training data sets are expected to be lower, thus the overall accuracy among resamples should be less variable. However, if migration progression depends on highly spatially and temporally variable predictors such as chlorophyll (food availability) then the variability in the accuracy among resamples is expected to be larger. We have now clarified this point in the text (see lines 481-485).

---

## [Decision Letter · Decision Letter 2]

13 Jul 2020

Citizen science for predicting spatio-temporal patterns in seabird abundance during migration

PONE-D-19-29580R2

Dear Dr. Martin,

We’re pleased to inform you that your manuscript has been judged scientifically suitable for publication and will be formally accepted for publication once it meets all outstanding technical requirements.

Kind regards,

Vitor Hugo Rodrigues Paiva

Academic Editor

PLOS ONE

Additional Editor Comments (optional):

Reviewers' comments:

Reviewer's Responses to Questions

**Comments to the Author**

1. If the authors have adequately addressed your comments raised in a previous round of review and you feel that this manuscript is now acceptable for publication, you may indicate that here to bypass the “Comments to the Author” section, enter your conflict of interest statement in the “Confidential to Editor” section, and submit your "Accept" recommendation.

Reviewer #2: All comments have been addressed

2. Is the manuscript technically sound, and do the data support the conclusions?

Reviewer #2: Yes

3. Has the statistical analysis been performed appropriately and rigorously? 

Reviewer #2: Yes

4. Have the authors made all data underlying the findings in their manuscript fully available?

Reviewer #2: Yes

5. Is the manuscript presented in an intelligible fashion and written in standard English?

Reviewer #2: Yes

6. Review Comments to the Author

Reviewer #2: I think authors addressed all my comments and elucidated some of my concerns on the methods, particularly, the use of Lat and Long in the models (which was my main concern) and further description of the techniques applied.

7. PLOS authors have the option to publish the peer review history of their article (what does this mean?). If published, this will include your full peer review and any attached files.

Reviewer #2: No

---

## [Editor Report · Acceptance letter]

21 Jul 2020

PONE-D-19-29580R2 

Citizen science for predicting spatio-temporal patterns in seabird abundance during migration 

Dear Dr. Martin:

I'm pleased to inform you that your manuscript has been deemed suitable for publication in PLOS ONE. Congratulations! Your manuscript is now with our production department. 

Kind regards, 

on behalf of

Dr. Vitor Hugo Rodrigues Paiva 

Academic Editor

PLOS ONE